# The IL-33/ST2 axis is protective against acute inflammation during the course of periodontitis

Anhao Liu [1,2], Mikihito Hayashi[1], Yujin Ohsugi[2], Sayaka Katagiri[2], Shizuo Akira[3], Takanori Iwata[2] & Tomoki Nakashima [4]

Periodontitis, which is induced by repeated bacterial invasion and the ensuing immune reactions that follow, is the leading cause of tooth loss. Periodontal tissue is comprised of four different components, each with potential role in pathogenesis, however, most studies on immune responses focus on gingival tissue. Here, we present a modified ligature-induced periodontitis model in male mice to analyze the pathogenesis, which captures the complexity of periodontal tissue. We find that the inflammatory response in the peri-root tissues and the expression of IL-6 and RANKL by Thy-1.2⁻ fibroblasts/stromal cells are prominent throughout the bone destruction phase, and present already at an early stage. The initiation phase is characterized by high levels of ST2 (encoded by *Il1rl1*) expression in the peri-root tissue, suggesting that the IL-33/ST2 axis is involved in the pathogenesis. Both *Il1rl1*- and *Il33*-deficient mice exhibit exacerbated bone loss in the acute phase of periodontitis, along with macrophage polarization towards a classically activated phenotype and increased neutrophil infiltration, indicating a protective role of the IL-33/ST2 axis in acute inflammation. Thus, our findings highlight the hidden role of the peri-root tissue and simultaneously advance our understanding of the etiology of periodontitis via implicating the IL-33/ST2 axis.

Periodontitis is the leading cause of tooth loss and affects nearly 19% of the global adult population, with an increasing incidence[1,2]. Inflammation caused by periodontitis not only destroys periodontal tissues, but also facilitates several systemic diseases that have potentially serious consequences[1,3,4]. Although the importance of oral hygiene in controlling this infection-associated disease has been generally acknowledged throughout history[5], it is challenging to render such hygiene sufficiently protective[6,7]. Therefore, investigations into the etiology of this disease have been undertaken in an effort to improve this situation, and it was found that various microbes, such as *Porphyromonas gingivalis* play a role in periodontitis[8–10]. However, the differences among the different species[11–13] and eras[14,15] have been

observed, and the susceptibility to periodontitis among the different populations and genetic backgrounds is also inconsistent[1,16]. As periodontitis is no longer simply the result of bacterial invasion[17,18], breakthroughs may arise by investigating the host immune response.

Various types of immune cells[19–21] and cytokines[22] are involved in the pathogenesis and tissue destruction in periodontitis. Certain products of the immune response, such as matrix metalloproteinases[23] have been implicated in this phenomenon, and it was recently shown that Foxp3⁺ T-cell-derived Th17 (exFoxp3⁺ Th17) cells are involved in the pathogenesis of experimental periodontitis[24,25]. Furthermore, single-cell RNA sequencing (scRNA-seq) studies have characterized immune cells that are active during the pathogenesis, furthering our

[1]Department of Cell Signaling, Graduate School of Medical and Dental Sciences, Tokyo Medical and Dental University, 1-5-45, Yushima, Bunkyo-ku, Tokyo 113-8549, Japan. [2]Department of Periodontology, Graduate School of Medical and Dental Sciences, Tokyo Medical and Dental University, 1-5-45, Yushima, Bunkyo-ku, Tokyo 113-8549, Japan. [3]Laboratory of Host Defense, IFReC,Osaka University, 3-1 Yamadaoka, Suita, Osaka 565-0871, Japan. [4]Faculty of Dentistry, Tokyo Medical and Dental University, 1-5-45, Yushima, Bunkyo-ku, Tokyo 113-8549, Japan. ✉e-mail: naka.csi@tmd.ac.jp

understanding of the mechanisms underlying periodontitis[26–28]. However, these studies are usually performed exclusively on gingival samples, although periodontal tissue is composed of the gingiva, periodontal ligament, alveolar bone, and cementum[29]. This sampling strategy limits the conclusions that may be drawn from these studies; methods that allow simultaneous analysis of all tissue components are needed.

Gingivitis is the first stage of periodontitis and its aggravation eventually leads to bone destruction. Epithelial cells act as a barrier and, together with immune cells, exert an effort to prevent this process[18]. Damage-associated molecular patterns (DAMP)[30] released from damaged or dead cells are known to be essential for periodontitis pathogenesis because they are involved in immune responses related to tissue destruction. However, among DAMPs, the interleukin (IL)-33/ST2 (encoded by *Il1rl1*) axis, which functions as a vital alarmin in various infectious diseases and is crucial for mucosal immunity, has not been extensively studied in periodontitis[31]. IL-33, unlike other members of the IL-1 superfamily, is continuously present in the nucleus of tissue component cells (especially epithelial cells), and released when cells reach the end of their life due to infection or trauma[32]. The IL-33 receptor is composed of the IL-1 receptor accessory protein (IL-1RAcP) and ST2, and ST2 not only acts as a binding site but also has a soluble decoy isoform (sST2, translated by different transcript variants from the same gene) that modulates signaling[31]. In various infectious and allergic diseases, the IL-33/ST2 axis has been implicated in transducing proinflammatory signals to immune cells via myeloid differentiation primary response 88 (MyD88)-related pathways. This signal mainly induces the activation of mast cells, type 2 innate lymphoid cells, Th2 cells, other granulocytes, and macrophages, which are responsible for allergy and type 2 immune reaction[33–36]. However, there is a possible diversity in the function of this axis in different organs and tissues, including the immunosuppressive, pro-fibrogenic, and tissue-regenerative effects in certain situations, especially in the case of inflammation control and tissue repair[33,37–39]. Although the IL-33/ST2 axis tends to be considered an alarmin in gingival infection[31], its exact role in the pathogenesis may vary owing to the diversity of this axis among diseases[40]. The function of the IL-33/ST2 axis in the pathogenesis of periodontitis remains unclear and requires further systematic investigation.

In this study, we develop a modified ligature-induced periodontitis model that provides a sufficient sample yield from individual mice, allowing tissue separation into gingival tissue (GT), peri-root tissue (PRT), and bone tissue (BT) for a more detailed analysis of the pathogenesis of periodontitis. Analysis of this model reveals the critical role of fibroblasts/stromal cells in the PRT for osteoclastogenesis and bone destruction by producing IL-6 and receptor activator of nuclear factor κ-B ligand (RANKL). Moreover, we find that the IL-33/ST2 axis protects against the acute inflammation in initiation phase by acting on the unique periodontal tissue-resident macrophages (PTRM). Thus, our results provide in vivo evidence for the critical role of the PRT and the unique character of the IL-33/ST2 axis in the pathogenetic process of periodontitis.

## Results

### The triple ligature method as a modified model of periodontitis

To improve the yield of RNA from individual samples compared to that with the classical single-ligature model[41], periodontitis was induced by triple ligature placement on the left upper molar of mice (Supplementary Fig. 1a). Bone resorption in the modified triple ligature model was compared with that in the conventional single ligature model using μCT and histological analysis based on the time course (Supplementary Fig. 1b). Bone resorption was initiated on day 5, peaked on day 8, and stabilized on day 14 in both of these methods, which can be defined as the initiation phase (IP), acute inflammation phase (AP), and chronic inflammation phase (CP) in the pathogenetic process.

Compared with the single ligature model, the triple ligature model exhibited a significant increase in total bone loss but a relatively mild enhancement of destruction around the second molar (Fig. 1a, b). However, the tartrate-resistant acid phosphatase (TRAP)-positive osteoclast number normalized to bone surface length in the second molar region was similar between the two models (Fig. 1c, Supplementary Fig. 1c). Immunofluorescence staining for matrix metalloproteinase 9 (MMP9) in the triple ligature model showed that the number of MMP9-positive cells on the bone surface, including osteoclasts, was similar to that of TRAP-positive cells. These results indicated that the modified model enhanced bone loss mainly by expanding its range (Supplementary Fig. 2a, b). To confirm the advantage of this model in the tissue-separated analysis of periodontitis, we further isolated three tissues and evaluated the RNA yield between the two models (Supplementary Fig. 3a). The results showed that the triple ligature model effectively increased the yield, achieving four times the yield of normal PRT and supporting the high-resolution analysis of different tissues (Supplementary Fig. 3b).

### Different roles of three tissues during the pathogenetic process

After confirming the distinct separation of the tissues with tissue-specific genes (Supplementary Fig. 3c), we investigated the roles of the three tissues in pathogenetic process by examining the expression of representative cytokines using the modified triple ligature model. Under physiological conditions, the mRNA expression of proinflammatory cytokines differed among the three tissues and was higher in the GT, except for *Il17a* (Fig. 1d, Supplementary Fig. 4a). Although the *Tnfsf11/Tnfrsf11b* ratio was similar among all tissues, the expression of *Tnfsf11* (which encodes RANKL), *Tnfrsf11b* (which encodes OPG), and *Il10* was significantly lower in the GT (Fig. 1d, Supplementary Fig. 4b, c). Next, we examined the sequential changes in cytokine expression and found that most of the proinflammatory cytokines were significantly increased in all tissues after ligature placement and decreased on day 3, while *Il17a* remained high in the GT (Fig. 1e, Supplementary Fig. 4d). Following this upregulation in the early stage, *Il6* and *Tnfsf11* markedly increased in the PRT in IP, but *Tnf* and *Il1b* showed no apparent changes after the gingivitis phase. With a slight increase in *Tnfrsf11b*, the *Tnfsf11/Tnfrsf11b* ratio was upregulated only in the PRT (Fig. 1e, Supplementary Fig. 4e), and continuous upregulation of *Il10* in the PRT and BT was detected at the end of IP to CP, especially with a bilateral increase in the BT (Supplementary Fig. 4f). These RT-qPCR results for proinflammatory cytokines were confirmed at the protein level by immunofluorescence staining (Supplementary Fig. 5). We also found that bone resorption was preceded by the cleavage of collagen fibers with a tendency from the GT to PRT, which was accompanied by the release of MMP9 by non-osteoclastic cells, indicating that the changes in the PRT occurred before osteoclasts function (Supplementary Fig. 2, and 6a, b). Furthermore, concurrent with the increased expression of *Il6* and *Tnfsf11* in the PRT, angiogenesis peaked in IP and subsequently increased in the GT after AP (Supplementary Fig. 6c, d), also indicating the importance of IP and the PRT in the pathogenesis.

### The localized and systemic changes in cellular components/cytokine sources in initiation phase

mRNA expression and histological analysis suggested that IP is critical for bone resorption, therefore, we examined the changes in cellular components and the source of proinflammatory cytokines at this time point using flow cytometry. We observed an increase in CD45$^+$ immune cells and MHCII$^+$ antigen-presenting cells in the GT and PRT, but a decrease in the BT, suggesting that the immune response to invading bacteria occurs mainly in the former (Supplementary Fig. 7a, b). In contrast to the intense increase in CD11b$^+$ myeloid cells in the GT and PRT, the restricted increase in CD3ε$^+$ T cells in the GT and the limited involvement of CD19$^+$ B cells indicate that the innate immune response was dominant in IP (Supplementary Fig. 7c–f).

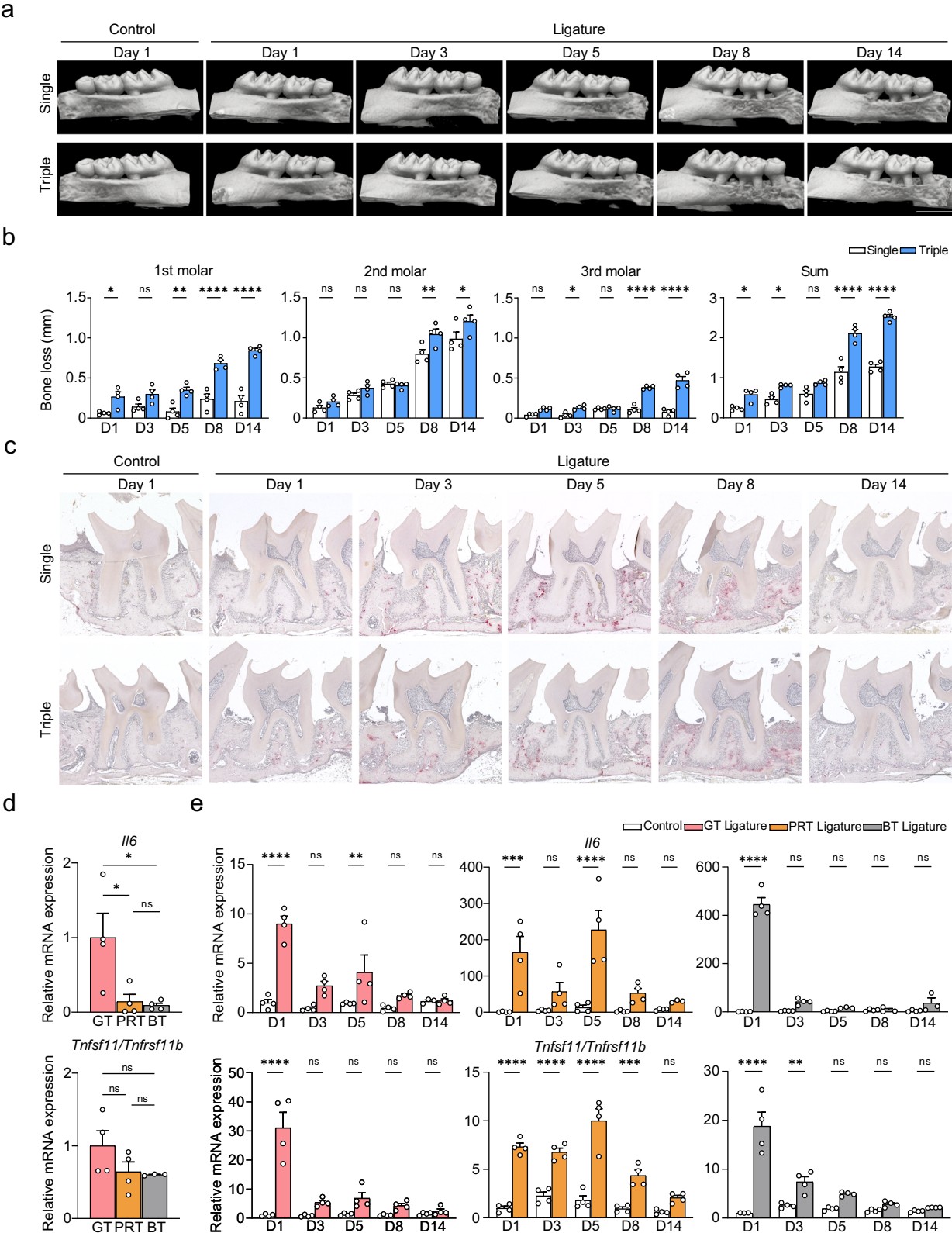

Based on the changes in the cellular components in IP, we investigated the source of cytokines in the GT and PRT and found that CD3ε⁺ T cells mainly produced IL-17A in the GT, in accord with previous studies (Supplementary Fig. 8). Although certain cell types exhibited increased production of IL-6 (Supplementary Fig. 9a, b), the major source in both tissues was local CD45⁻ tissue component cells (Fig. 2a–c). Consistent with the results of RT-qPCR (Fig. 1e), the

changes in IL-6 were mild in the GT, but intense in the PRT (Fig. 2a, Supplementary Fig. 9a, b), where it was primarily released from Thy-1.2⁻ fibroblasts/stromal cells (Supplementary Fig. 9c). We then evaluated RANKL production and found that the cells expressing RANKL were CD3ε⁺ T cells and CD45⁻ tissue-component cells in the GT and PRT, respectively, suggesting different responses during IP (Fig. 2d–f, Supplementary Fig. 10). Collectively, considering the RT-qPCR results

**Fig. 1 | Comparison of the two types of ligature models and temporal mRNA expression profiling of cytokines. a** Representative maxillary μCT images. Scale bar, 1 mm. **b** Temporal change of bone loss on different tooth regions. From left to right: the first molar, the second molar, the third molar, and their sum ($n = 4$ mice per group). **c** Representative TRAP staining images. Scale bar, 300 μm. **d** mRNA expression of *Il6* and the *Tnfsf11/Tnfrsf11b* ratio on the control side of the three tissues (on Day 1; GT = 1; $n = 4$ mice per group except for the group highlighted in Source Data file, which was $n = 3$ mice). **e** Temporal changes in mRNA expression of *Il6* and the *Tnfsf11/Tnfrsf11b* ratio in the three tissues (control Day 1 of each tissue = 1; $n = 4$ mice per group except for the groups highlighted in Source Data file, which were $n = 3$ mice per group). GT gingival tissue, PRT peri-root tissue, BT bone tissue, D1 Day 1, D3 Day 3, D5 Day 5, D8 Day 8, D14 Day 14. Data are presented as the mean ± SEM. *$P < 0.05$; **$P < 0.01$; ***$P < 0.001$; ****$P < 0.0001$; ns (not significant), $P > 0.05$; by one-way ANOVA with multiple comparisons via Tukey's test (**d**); and two-way ANOVA with multiple comparisons via Šídák's method (**b, e**). The obvious outliers were evaluated and excluded by Grubb's test ($\alpha = 0.05$). Exact $P$ values are presented in Supplementary Data. 1. Source data are provided as a Source Data file.

showing that PRT was the dominant source of RANKL in IP (Supplementary Fig. 4b, e), Thy-1.2⁻ fibroblasts/stromal cells were the major producers of RANKL.

Similar experiments were performed on blood specimens in IP to further examine the local and systemic relationship in the pathogenesis of periodontitis and to provide a rationale for the exploring the mechanism of periodontitis. However, we found that CD19⁺ B cells increased, while CD3ε⁺ T cells unexpectedly decreased, and CD11b⁺ myeloid cells had no noticeable change (Supplementary Fig. 11a). Few cells were able to produce IL-6 and IL-17A in the circulation (Supplementary Fig. 11b, c), but RANKL was produced more by CD3ε⁺ T cells and significantly decreased in IP. These results indicate that the local production of proinflammatory cytokines and the involvement of tissue-component cells are more important for pathogenesis and they do not induce a systemic inflammatory state as previously considered, at least for T cells and myeloid lineage cells in IP (Fig. 2g, h, Supplementary Fig. 11d). Given the biphasic upregulation of *Il10* expression in the BT and the fact that it has the richest blood supply among the three tissues (Supplementary Fig. 4f), we hypothesized and verified that IL-10-producing cells are present in the circulation, making the inflammatory state immunosuppressive (Fig. 2i and Supplementary Fig. 12a). IL-10-producing cells increased, and staining for surface antigens revealed that the major source was CD19⁺ B cells (Fig. 2j), characterized as B10[42] or M-B[43] cells. Since *Il10* expression was increased in the PRT in AP prior to the BT, we tested its local sources and found that the CD45⁻ and CD11b⁺ cells were the primary source, but in low absolute numbers, especially CD45⁻ cells (Supplementary Fig. 12b–d), indicating differences between local and systemic immunosuppressive processes.

### Involvement of the IL-33/ST2 axis in the pathogenetic process of periodontitis as revealed by comprehensive analysis

In addition to providing a brief overview of the pathogenesis of periodontitis, we performed bulk RNA-seq of the GT, PRT, and BT in IP to analyze gene expression more comprehensively, focusing on the initiation of the pathogenetic process. We detected 1022, 811, and 313 upregulated differentially expressed genes (DEG) and 353, 1221, and 369 downregulated DEGs in the three tissues, respectively (Supplementary Fig. 13a). Gene ontology (GO) term analysis of the upregulated DEGs revealed an enrichment of terms related to leukocyte migration and proliferation in the GT and PRT (Supplementary Fig. 13b), whereas extracellular matrix organization was enriched in the BT. Meanwhile, Kyoto Encyclopedia of Genes and Genomes (KEGG) pathway analysis of the upregulated DEGs indicated that the "cytokine-cytokine receptor interaction" was the most enriched pathway among the three tissues, as expected (Fig. 3a, Supplementary Fig. 13c). Furthermore, consistent with previous reports[24], the "IL-17 signaling pathway" was enriched at different levels in all three tissues, but "Th17 differentiation" was enriched only in the GT, indicating that the Th17 located in the GT is involved in the acceleration or initiation of pathogenesis. Notably, "Rheumatoid arthritis" and other systemic diseases considered to be related to periodontitis were enriched in the PRT, suggesting the potential of the PRT in these diseases and periodontal medicine. We performed Seq-immuCC[44] analysis on the data to predict changes in immune cell components and found that they differed among the three tissues. Differences in specific components between tissues were found mainly in neutrophils, macrophages, and B cells, with a marked increase in macrophage-related gene expression in the PRT (Supplementary Fig. 13d). Neutrophil-related gene expression was maintained in the BT and less frequently observed in the GT or PRT in IP.

To identify the genes important for the periodontitis pathogenesis, we plotted the DEGs of the three tissues on a volcano plot and found that *Il1rl1* was significantly altered in the PRT (Fig. 3b, Supplementary Fig. 13e). After confirming the changes and trends in *Il1rl1* expression between tissues using RT-qPCR (Fig. 3c, d), we tested the expression of the major transcript variants (variant 1 encodes mST2 and variants 2 encodes sST2) and found that the trends and fold changes of variants 1/2 were similar to those of the total transcripts (Supplementary Fig. 14a, b). Considering that the change in variant 1 was mild (Fig. 3e, f), variant 2 accounted for the majority of the transcripts. These results indicate that the PRT mainly upregulates sST2 to block the IL-33/ST2 signaling pathway in IP; however, there is still the potential for sST2 to have additional functions.

The specific ligand for ST2 is IL-33, therefore, we investigated the expression of *Il33* along with other interleukins in the RNA-seq data. *Il33* expression significantly increased in the PRT and BT in IP, but not in the GT (Supplementary Fig. 14c). In contrast to the RNA-seq results, RT-qPCR analysis revealed that *Il33* expression was significantly higher in the GT under normal conditions, with a notable increase before IP (Fig. 3g, h). *Il33* expression had a peak in the GT during the gingivitis phase (day 1-3), followed by an increase in the PRT after IP, showing a trend consistent with the time course of bone resorption (Fig. 1a, b, 3g, h). To elucidate the reasons for the discrepancy between the two results, we examined the expression of major transcript variants of *Il33* (variants 1 and 2)[45]. Although these variants encode the same protein, the results showed that transcript variant 2 caused differences in expression between tissues and was the major variant in the GT. The expression of transcript variant 1 was comparable between normal tissues, significantly upregulated after ligature placement, and persistently elevated in the PRT after IP, with the major expression in the GT (Supplementary Fig. 14d–g).

### The source and distribution of IL-33 and ST2 in IP

To gain further insight into IL-33/ST2-mediated pathogenesis of periodontitis, we examined cells expressing ST2 and IL-33 in the GT and PRT using flow cytometry. The major population of ST2-expressing cells in the PRT were CD45⁻ tissue component cells, indicating that they were the primary source of sST2 (Fig. 4a, Supplementary Fig. 15). As for the IL-33-producing cells, CD45⁻ tissue component cells were the primary source, whereas CD45⁺ immune cells were slightly increased in the two tissues (Fig. 4b, Supplementary Fig. 16). We examined the composition of ST2⁺CD45⁻ and IL-33⁺CD45⁻ cells in the two tissues and found that the primary source of ST2 in the GT was Thy-1.2⁺ fibroblasts/stromal cells, whereas that in the PRT was Thy-1.2⁻ fibroblasts/stromal cells (Fig. 4c, e). IL-33 was mainly detected in EpCAM⁺ epithelial cells in the GT and Thy-1.2⁻ fibroblasts/stromal cells in the PRT (Fig. 4d, f). Based on the results of RT-qPCR, RNA-seq, and cell number changes, it is likely that the Thy-1.2⁻ fibroblasts/stromal cells in the PRT are the predominant source of sST2 during pathogenesis

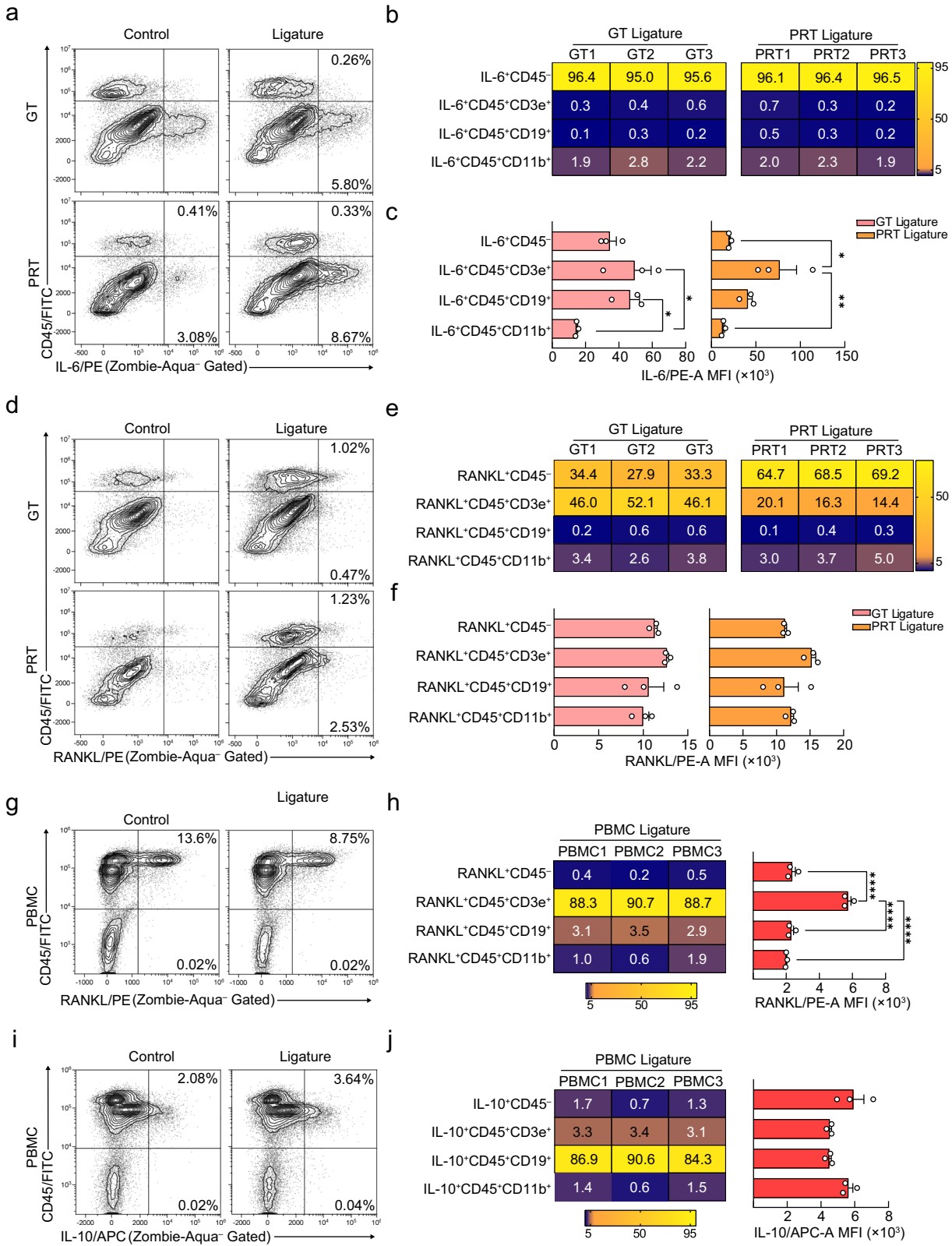

(Fig. 3c–f, 4a, Supplementary Fig. 15a–c). For IL-33, however, there are multiple possible sources, as two transcript variants exist, and their expression levels vary over time in the different tissues (Supplementary Fig. 14d–g).

To confirm these results, we performed immunofluorescence staining for ST2 and IL-33 along with cell markers of the major cell populations identified using flow cytometry. In the intact

periodontal tissue, ST2 expression was weak and only observed in a small number of CD45$^+$ and Thy-1.2$^-$ cells (Fig. 4g, Supplementary Fig. 17). When periodontitis was induced, ST2$^+$Thy-1.2$^+$ and ST2$^+$Thy-1.2$^-$ fibroblasts/stromal cells appeared in the GT and PRT, which was consistent with the flow cytometry results (Fig. 4g, Supplementary Fig. 17). IL-33$^+$ cells were found in the PRT of the second molar region, but few signals were found in the GT and no signal was found in

**Fig. 2 | Verification of the local and systemic source of inflammation-related cytokines in initiation phase. a**, **d** Changes in IL-6$^+$ (**a**) and RANKL$^+$ (**d**) cells in the GT and PRT with or without ligature placement. A representative contour plot is shown, and the percentages presented in the gates are the mean of three independent mouse experiments. **b**, **e** Heat map of the positive percentage of different IL-6$^+$ (**b**) and RANKL$^+$ (**e**) lineages in the ligated GT and PRT. The data from three independent experiments are shown ($n$ = 3 mice per group, GT/PRT 1, 2, 3). **c**, **f** MFI (median fluorescence intensity) of different levels of IL-6$^+$ (**c**) and RANKL$^+$ (**f**) in the ligated GT and PRT ($n$ = 3 mice per group). **g**, **i** Changes in RANKL$^+$ (**g**) and IL-10$^+$ (**i**) cells in the PBMC with or without ligature placement. The percentages shown in the gates are the mean of three independent mouse experiments, and a representative contour plot is shown. **h**, **j** Left: Heat map of the positive percentage of different RANKL$^+$ (**h**) and IL-10$^+$ (**j**) lineages in the ligated PBMC. Data from three independent experiments are shown ($n$ = 3 mice per group, PBMC 1, 2, 3). Right: MFI of the different RANKL$^+$ (**h**) and IL-10$^+$ (**j**) lineage in the ligated PBMC ($n$ = 3 mice per group). GT gingival tissue, PRT peri-root tissue, PBMC Peripheral blood mononuclear cell. Data are presented as the mean ± SEM except for (**a**, **d**, **g**, **i**). *$P$ < 0.05; **$P$ < 0.01; ***$P$ < 0.001; ****$P$ < 0.0001; ns (not significant), $P$ > 0.05; by one-way ANOVA with multiple comparisons via Tukey's test (**c**, **f**, **j**). The related gating strategy was presented in Supplementary Fig. 29. Exact $P$ values are presented in Supplementary Data. 1. Source data are provided as a Source Data file.

EpCAM$^+$ gingival sulcular epithelial cells when the periodontal tissue was intact (Fig. 4h, Supplementary Fig. 18). We attempted to resolve this discrepancy by searching for IL-33-positive cells in the gingival tissue outside of the second molar region and found that IL-33$^{high}$ cells were located in the submucosal layer of the mesial gingiva of the first molar, which was far from the region of the tooth (Supplementary Fig. 19a). These "IL-33-enriched submucosal stromal cells (IL-33SC)" were CD45$^-$, tended to be Thy-1.2$^+$ where they approached the tooth, showed high IL-33 signals on and around their nucleus, as in nasal epithelial cells, and notably had an EpCAM$^-$ gingival epithelium as a superstratum (Supplementary Fig. 19). We then examined the expression of IL-33 in periodontal tissues after the induction of periodontitis. IL-33$^+$EpCAM$^+$ gingival sulcular epithelial cells (IL-33EC) were observed in the GT, whereas a reduction in IL-33-positive CD45$^-$Thy-1.2$^-$ periodontal ligament cells (IL-33PC) was observed in the PRT (Fig. 4h, Supplementary Fig. 18). The intracellular location of IL-33 in IL-33ECs was perinuclear as in IL-33SCs, but in IL-33PCs it was clearly outside the nucleus (Supplementary Fig. 19b).

### The IL-33/ST2 axis plays a protective role in initiation phase by affecting the function of tissue-resident macrophages under inflammatory conditions

The complexity of IL-33-expressing cells and isoforms of ST2 limits a detailed analysis of their roles in wild-type (WT) mice. Therefore, we induced periodontitis in both $Il33^{\Delta/\Delta}$ (generated by crossing $Il33^{flox/flox}$ mice with $Actb$-$Cre$ mice; Supplementary Fig. 20a, b) and $Il1rl1^{-/-}$ mice to determine whether signaling of the IL-33/ST2 axis or its modulation by a soluble decoy contributes to pathogenesis. Notably, bone destruction in both strains was more severe in AP, but comparable in CP compared with that to WT mice. Although the cells in the PRT express high levels of sST2, the similar phenotypes observed in both $Il33^{\Delta/\Delta}$ and $Il1rl1^{-/-}$ mice suggest that accelerated bone destruction is due to the lack of IL-33/ST2 signaling and that its function is protective against local acute inflammation in periodontitis (Fig. 5a, b, Supplementary Fig. 21a). However, histomorphometric analysis showed only slight changes in osteoclast numbers compared to those in control mice (Fig. 5c, d), indicating an insignificant effect of the axis on osteoclastogenesis. Therefore, we examined cytokine expression patterns in IP and AP to elucidate the mechanisms by which the IL-33/ST2 axis promotes osteoclast function but does not affect osteoclastogenesis, as shown by the increased eroded surface in AP in both knockout mice (Fig. 5c, d). Compared with the mild changes in the $Tnfsf11/Tnfrsf11b$ ratio, $Il6$ expression in IP was significantly elevated in the periodontal tissues of both deficient mouse strains, especially in the PRT (Fig. 5e, f, Supplementary Fig. 21b–d). Flow cytometry analysis revealed an increased percentage of CD45$^+$ IL-6-producing cells, most of which were CD11b$^+$MHCII$^+$ myeloid cells (Fig. 5g, Supplementary Fig. 22), suggesting that this axis mainly affects the inflammatory conditions that enhance osteoclast function. Furthermore, the rapid recovery of damaged bone after AP in both kockout mice was also important, therefore, we analyzed $Il33^{\Delta/\Delta}$ mice by bulk RNA-seq (Fig. 5a, b). Pathway analysis between WT and $Il33$-deficient mice on day 11 (the process from AP to CP) showed an enrichment of

suppression of pathways associated with immune reactions in the PRT and upregulation of pathways associated with regeneration in the BT (Supplementary Fig. 23a–c, Supplementary Table 1–3). These data also showed the suppression of key cytokines, including $Il10$, in both strains, indicating the presence of potent local or systemic immunosuppressive mechanisms that are not solely dependent on IL-10 (Supplementary Fig. 23d). IL-10 is essential for the attenuation of inflammatory responses, and the low $Il10$ expression on day 8 in the BT of both deficient mice also partially supports this possibility (Supplementary Fig. 21e).

As the characterization of effector cells of the IL-33/ST2 axis is key to clarifying the mechanism of the protective effect, we carefully evaluated mST2-expressing cells in WT mice and found that most were CD11b$^+$MHCII$^+$ myeloid cells, while a small number of CD11b$^+$MHCII$^-$ myeloid or CD3ε$^+$ T cells were also presented in both tissues (Fig. 6a, b, Supplementary Fig. 24). The percentage of mST2$^+$CD11b$^+$MHCII$^+$ myeloid cells did not significantly change in the PRT, but increased in the GT after induction of periodontitis. Considering that the absolute number of these cells was higher in the intact PRT, it is likely that this population was mainly composed of tissue-resident immune cells in the PRT, such as tissue-resident macrophages (Fig. 6b, Supplementary Fig. 24a, b). To confirm the cell types of mST2$^+$ myeloid lineage populations according to the literature, we applied gating strategies to distinguish the major antigen-presenting myeloid cell subsets (APCMC, including dendritic cells, and M1/M2 macrophages), non-antigen-presenting myeloid cell subsets (non-APCMC, including monocytes, neutrophils, and mast cells), and helper and regulatory T cell subsets (including Th1, Th2, Th17, and Treg; details are documented in the Reporting Summary). Notably, we clarified that the CD11b$^+$MHCII$^+$CD11c$^-$CD86$^-$CD206$^+$ macrophage subset (defined as PTRM) is a major antigen-presenting cell population, based on previous results showing that APCMCs are abundant in periodontal tissue (Fig. 6c, Supplementary Fig. 24, 25a). Dendritic cells and M1/M2 macrophages also had mST2$^+$ subpopulations, but in relatively small numbers compared with PTRMs (Fig. 6c). In contrast, mST2$^+$ non-APCMCs were also present, albeit in low absolute numbers, and were relatively enriched in mast cells (Supplementary Fig. 25b, c). We also attempted to determine the subpopulation of mST2$^+$ T cells, but this was difficult because of their scarcity in the local helper T cell populations (Supplementary Fig. 25d, e).

After determining the effector cell types, we evaluated the changes in various cellular components between WT and IL-33- or ST2-deficient mice to explore the possible mechanisms underlying the phenotypes of these mice. Both mouse strains showed a trend toward a decrease in dendritic cells and macrophages; in particular, PTRMs and M2 macrophages were decreased, resulting in an increased M1/M2 macrophage ratio in both the GT and PRT, that is, a polarization shift of the macrophage population (Fig. 6d, e, Supplementary Fig. 26). Meanwhile, the CD11b$^+$MHCII$^-$CD115$^+$ monocyte population increased in $Il1rl1^{-/-}$ mice, suggesting that the loss of IL-33/ST2 signaling may affect monocyte-macrophage differentiation (Supplementary Fig. 27a, b). Notably, we observed a reversal in the proportion of MHCII$^+$ cells in the CD11b$^+$ myeloid cell lineage, which was caused by neutrophils, showing an

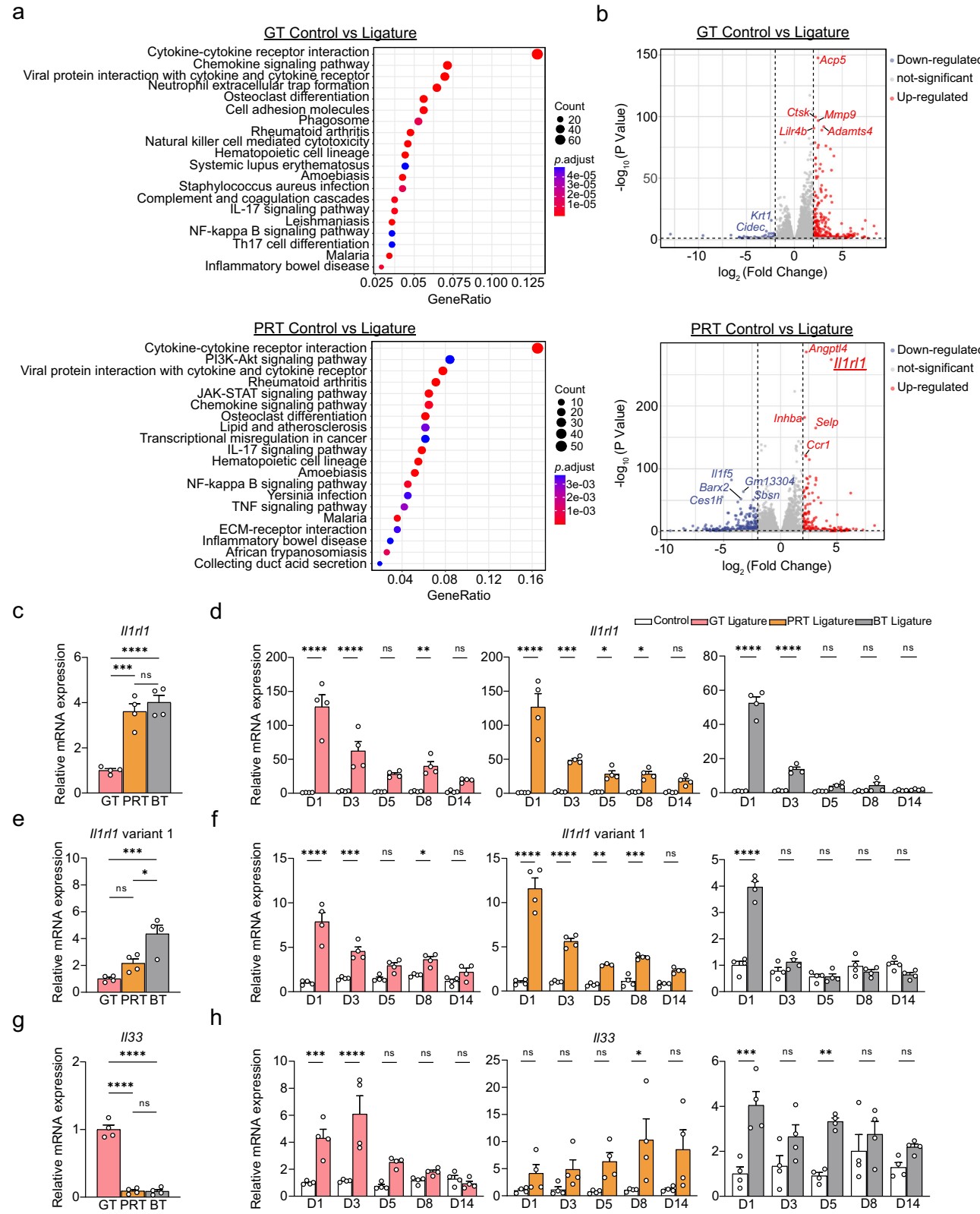

almost two-fold increase in both mouse strains (Fig. 6d, f, Supplementary Figs. 26, 27a, b). For other granulocytes, with the exception of neutrophils, mast cells were decreased in *Il1rl1⁻/⁻* mice and *Il33⁻/⁻* mice, suggesting that this axis is required for their differentiation and/or activation (Fig. 6f, Supplementary Fig. 27a, b). In addition, we investigated the specific sources of increased IL-6 levels and found that the majority of them were macrophages (Supplementary Fig. 27c).

Increased percentages of non-antigen-presenting cells also produced a certain amount of IL-6 and were relatively rich in neutrophils (Supplementary Fig. 27d).

To further investigate the direct effects of IL-33/ST2 signaling on macrophages, we assessed the effects of IL-33 on macrophage polarization in vitro. In typically polarized macrophages, the expression of mST2 (*Il1rl1* variant 1) was predominantly increased in the M2a subtype

**Fig. 3 | Comprehensive analysis of three tissues in the initiation phase and the discovery of the involvement of the IL-33/ST2 axis. a** KEGG pathway enrichment analysis based on the upregulated DEGs in the GT and PRT using GO/KEGG enrichment analysis tools based on clusterProfiler in Hiplot (Benjamini & Hochberg-method adjusted *P* value < 0.05; false discovery rate < 0.1; fold-change of normalized counts ≥ 2). The top 20 pathways are shown with their expression counts and adjusted (Benjamini & Hochberg method) *P* value. **b** Volcano plots of DEGs in the GT and PRT. The top 5 genes with the highest adjusted *P* value were annotated. DEG differentially expressed gene. **c, e, g** mRNA expression level of *Il1rl1* (**c**), *Il1rl1* variant 1 (**e**), and *Il33* (**g**) on the control side of three tissues (on Day 1; GT = 1; *n* = 4 mice per group except for the groups highlighted in Source Data file,

which were *n* = 3 mice per group). **d, f, h** Temporal change in the mRNA expression of *Il1rl1* (**d**), *Il1rl1* variant 1 (**f**), and *Il33* (**h**) in the three tissues (control Day 1 of each tissue = 1; *n* = 4 mice per group except for the groups highlighted in Source Data file, which were *n* = 3 mice per group). GT gingival tissue, PRT peri-root tissue, BT bone tissue, D1 Day 1, D3 Day 3, D5 Day 5, D8 Day 8, D14 Day 14. Data are presented as the mean ± SEM. \**P* < 0.05; \*\**P* < 0.01; \*\*\**P* < 0.001; \*\*\*\**P* < 0.0001; ns (not significant), *P* > 0.05; by one-way ANOVA with multiple comparisons via Tukey's test (**c, e, g**); and two-way ANOVA with multiple comparisons via Šídák's method (**d, f, h**). The obvious outliers were evaluated and excluded by Grubb's test (α = 0.05). Exact *P* values are presented in Supplementary Data. 1. Source data are provided as a Source Data file.

(Supplementary Fig. 28a). Importantly, pretreatment and stimulation with IL-33 had the ability to promote and inhibit the polarization of M2a and M1 macrophages, respectively (Fig. 6g, Supplementary Fig. 28b), leading to a shift toward M2. Consistent with the reduction in the signal that promotes M2 polarization in vivo in *Il1rl1*[-/-] mice, the effect of IL-33 treatment was abolished by the induction of polarization in bone marrow-derived macrophages (BMDM) from *Il1rl1*[-/-] mice (Fig. 6g, Supplementary Fig. 28b). In addition, IL-33 had little impact on M2c polarization, as shown by the mRNA expression of M2c markers, including *Ccr2* and *Mmp8*. However, IL-33 treatment significantly increased the expression of these genes in M0 macrophages, suggesting the presence of an M2 subtype associated with IL-33 (Supplementary Fig. 28b). We attempted to verify these effects at the protein level using flow cytometry. We found that IL-33 treatment significantly increased CD206 expression in M2a-polarized macrophages and slightly increased it in M2c-polarized macrophages (Supplementary Fig. 28c, d). This suggests a relationship between IL-33/ST2, CD206, and IL-4 signaling. Together, the in vivo and in vitro results demonstrated that PTRMs predominantly express mST2 within periodontal tissues and IL-33/ST2 signaling is necessary to regulate their functions during the inflammatory process, particularly M2a polarization. Defects in IL-33/ST2 signaling may affect not only macrophage differentiation but also well-controlled immune responses, such as neutrophil chemotaxis (Fig. 6f, Supplementary Fig. 27a, b).

## Discussion

Due to the ethical constraints of human research, much of the knowledge on periodontitis has been obtained from mouse studies using various disease models. From the 1990s to 2010, the primary method was to inject lipopolysaccharides into mice or administer plaques and bacteria[46,47]. Although the consensus in bacteriology is constantly changing[8–10], a ligature model using the host microbiota has been modified for mice[41]. The advantages of this model have been confirmed[48], and it is becoming a major method for inducing experimental periodontitis[49]. However, the inflammatory region in this model is restricted to the second molar because of the single-tooth ligature placement, making tissue collection difficult for sufficient sample yield and inaccurate tissue separation. In this study, we developed a modified mouse periodontitis model with a higher sample yield (Supplementary Fig. 3b). This enabled the high-resolution analysis of the periodontal tissues, which was difficult using previous models. To improve the resolution at the protein level as well, we combined Kawamoto's method with a tyramide signal amplification system to enable stable spatial detection of low-expressed proteins in vivo, facilitating further spatiotemporal analysis of hard tissues. Furthermore, we optimized the tissue separation and digestion methods and demonstrated their separation accuracy and rational cellular composition using RT-qPCR and flow cytometry (Supplementary Fig. 3c, 7). The modified model is more reliable for downstream experiments such as scRNA-seq, which requires specific parts or the complete cellular components of the periodontal tissues. These methods and tissue separation concepts could be used flexibly and alternatively in human studies and will improve our

understanding of the pathogenesis of periodontitis with a higher level of evidence.

By combining the results of RT-qPCR, flow cytometry, and immunofluorescence staining, we showed the changes in the cellular composition during the development of periodontitis and revealed the key functions of periodontal ligament tissues. Contrary to traditional concepts, considering that *Il1b* and *Tnf* are crucial in inflammatory diseases, including periodontitis, we observed no significant changes in their expression in IP and AP (Supplementary Fig. 4d, 5), and showed that the Thy-1.2⁻ fibroblast/stromal cell-derived RANKL and IL-6 in the PRT was more pronounced. Owing to the crucial roles of the RANKL/OPG system in osteoclastogenesis[50,51], the mRNA expression of *Tnfsf11* and *Tnfrsf11b* is typically used to evaluate trends in bone resorption. The results of higher *Tnfsf11* expression in the PRT than in the GT suggest that the results from studies using only gingiva may be insufficient and require further verification. Additionally, because the expression of osteoclastogenic cytokines was markedly upregulated in the Thy-1.2⁻ fibroblasts/stromal cells of the PRT during IP, it is plausible that Thy-1.2⁻ fibroblasts/stromal cells in the PRT are a key population in periodontitis. Although we mainly focused on the role of the PRT in bone destruction, the PRT may also have an immunosuppressive function under physiological conditions and partially during the post-IP period (Supplementary Fig. 4b, c, e, f, 12b, d). The immunosuppression may ultimately be systemic (Fig. 2g, i, Supplementary Fig. 4f); the expression of *Il10* and *Tnfrsf11b* in the PRT suggests the possibility that the PRT acts in a regulatory manner in periodontal tissues, although this requires further investigation. Furthermore, we emphasize that our results do not imply that the GT is unnecessary or unimportant, as the changes in the PRT are based on inflammation in the GT. Research on the GT is invaluable as well and provides a better understanding of periodontitis together.

Optimization of the tissue collection method improved the quality of the RNA-seq analysis of the isolated samples, resulting in the discovery of the importance of the IL-33/ST2 axis in periodontitis after confirming the significant upregulation of *Il1rl1* in the PRT. In contrast to previous studies[52–54], we demonstrated the protective function of the IL-33/ST2 axis and its suppressive effect on *Il6* expression in IP by using *Il1rl1*[-/-] and *Il33*[Δ/Δ] mice. These results were confirmed using flow cytometry, with mST2⁺ PTRMs being the primary effector cells in IP. Abnormalities in these cells shifted the polarization of activated macrophages to an M1 phenotype by reducing the signal associated with polarization to M2 macrophages, increasing IL-6 production, and enhancing the migration of phagocytes, such as neutrophils. In two previous representative studies reporting that IL-33 exacerbates inflammation in periodontitis, one was performed by intraperitoneal administration of recombinant murine IL-33 before and after oral infection with *Porphyromonas gingivalis* in WT mice, and then analyzed 2 weeks later[52]; and the other was induced periodontitis in the same way using *Il1rl1*[-/-] mice[55]. Although both studies showed the exacerbating effect of IL-33 in different ways, the differences from our study are presumably mainly due to the systemic effects of extra IL-33, short induction time, and the reliability of bone resorption[56]. In vitro, IL-33 has also been

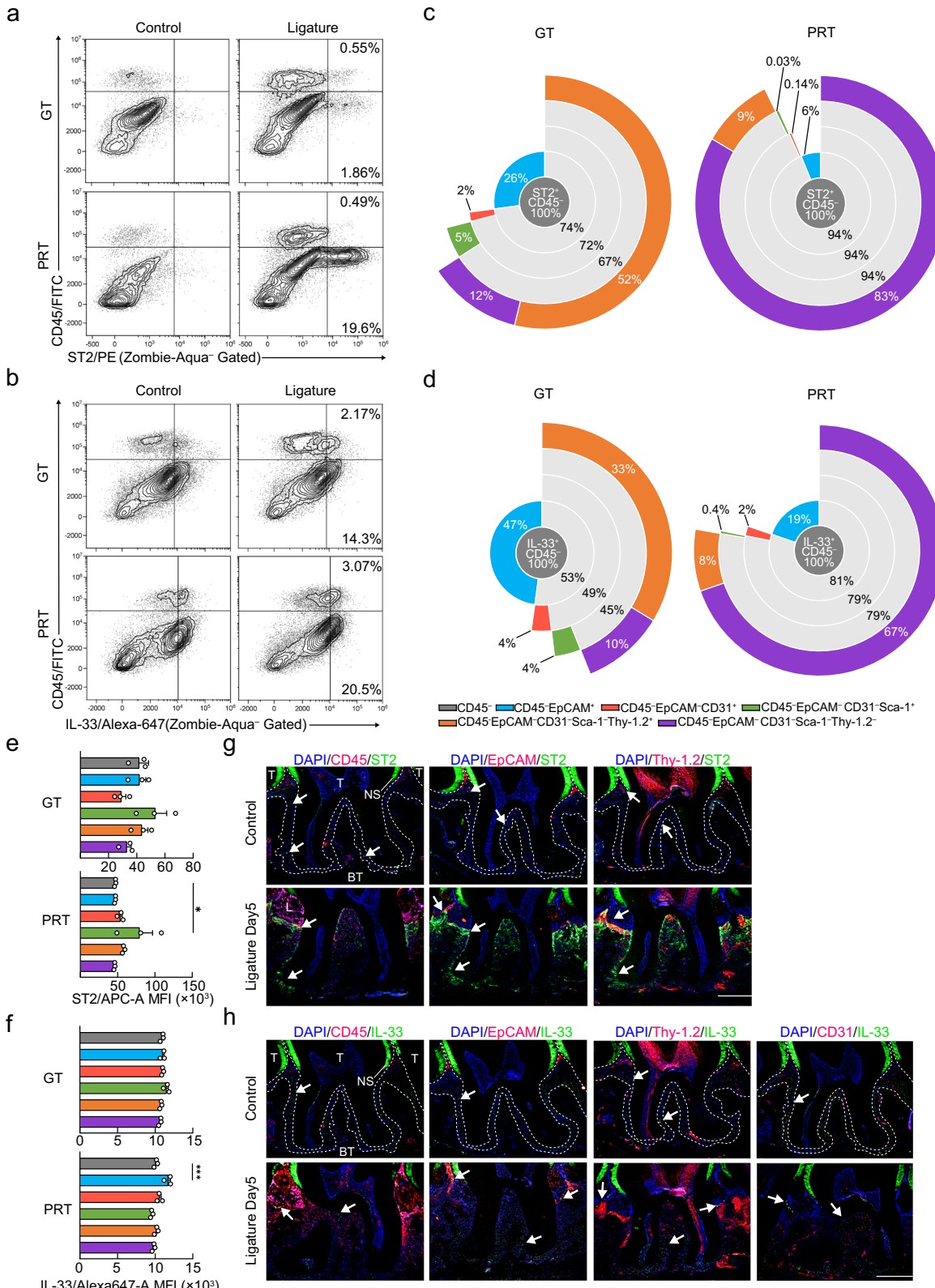

reported to inhibit TNF-α-induced osteoclastogenesis[57]. However, our results suggest that osteoclastogenesis in periodontitis is mainly dependent on RANKL and IL-6, as the expression of *Tnfsf11* and *Il6* was markedly increased, whereas the change in *Tnf* expression was mild. It is plausible that osteoclastogenesis was unaffected in *Il1rl1*[−/−] and *Il33*[Δ/Δ] mice, and that the axis acted more on inflammatory and other cells rather than directly on osteoclasts. Although further

direct evidence (e.g., induction of periodontitis in mST2-specific knockout mice) is needed, our findings shed some light on the involvement of the IL-33/ST2 axis in the pathogenesis of experimental periodontitis. Abnormalities in this axis may affect the function of tissue-resident macrophages and alter their polarization or migration, leading to hyperactive tissue inflammation and severe bone destruction during IP to AP (Fig. 7b).

**Fig. 4 | Verification of the source of ST2 and IL-33 in initiation phase.**
**a**, **b** Changes in ST2⁺ (**a**) and IL-33⁺ (**b**) cells in the GT and PRT with or without ligature placement. A representative contour plot is shown, and the percentages in the gates are the mean of three independent mouse experiments. **c**, **d** Sunburst map showing the composition of ST2⁺CD45⁻ (**c**) and IL-33⁺CD45⁻ (**d**) cells in the GT and PRT. The percentages of the populations shown are the mean of three independent mouse experiments. **e**, **f** MFI (Median fluorescence intensity) of the different ST2⁺CD45⁻ (**e**) and IL-33⁺CD45⁻ (**f**) lineages in the ligated GT and PRT ($n = 3$ mice per group). **g**, **h** Representative immunofluorescence staining images of three independent experiments of ST2 (**g**) and IL-33 (**h**) are shown with membrane marker proteins in the second molar region. Scale bar, 300 μm. T Tooth, NS non-specific signal, GT gingival tissue, PRT peri-root tissue, BT bone tissue. Data are presented as the mean ± SEM for (**e**, **f**). *$P < 0.05$; **$P < 0.01$; ***$P < 0.001$; ****$P < 0.0001$; ns (not significant), $P > 0.05$; by one-way ANOVA with multiple comparisons via Dunnett's test (**e**, **f**). The related gating strategy is shown in Supplementary Fig. 29 (**a**, **b**), 30 (**c**–**f**). Exact $P$ values are presented in Supplementary Data 1. Source data are provided as a Source Data file.

Immunofluorescence staining also revealed the complicated and diverse roles and sources of IL-33 in a tissue. Of the three types of IL-33-producing cells around the periodontal tissue, IL-33 was encoded by transcript variant 2 in IL-33SCs when the periodontal tissue was normal. The expression profile suggests that IL-33 acts mainly as an alarmin, released when stimulated by infection or tissue damage, and may be involved in the production of sST2, an antagonist of IL-33/ST2 signaling, from Thy-1.2⁻ fibroblast/stromal cells in the PRT. The predominant expression of IL-33 encoded by variant 2 in IL-33SCs made the GT enriched for *Il33* expression, and epithelial cells outside the papillary region showed different characteristics. However, the absence of the IL-33/ST2 axis does not affect the bone destruction in periodontitis, suggesting that at least the action of the IL-33/ST2 axis as an alarmin is not critical in the pathogenesis of periodontitis. In contrast, IL-33 produced by IL-33PCs and IL-33ECs was mainly encoded by transcript variant 1, according to the RT-qPCR results. Since the absence of IL-33 did not result in macroscopic changes in intact periodontal tissues of young adult male mice, it can be concluded that IL-33 produced by IL-33PCs is also not closely related to the periodontitis. Therefore, it is possible that IL-33 produced by IL-33ECs play a vital role in the pathogenesis. IL-33 released from IL-33ECs is highly expressed after induction of periodontitis, but tends to be stable after the acute phase, suggesting that its function as an alarmin is not important. Its main action seems to be to control the immune reaction, making inflammation less severe. The reduced expression of IL-33 in IL-33ECs after the tissue repair and the absence of IL-33SCs under the gingival epithelium may also explain the difficulty in detecting IL-33 in gingival crevicular fluid[58,59]. Similar epithelium-macrophage crosstalk has been observed in studies aimed at epithelial regeneration of alveoli[38], indicating that this mechanism may be present in other types of epithelial tissues and macrophages. The large amount of sST2 produced by Thy-1.2⁻ fibroblast/stromal cells in the PRT could be more easily transported throughout the body than bacterial components or immune cells. This sST2 may affect other organs or tissues and induce disease by blocking local IL-33/ST2 signaling, and is likely to play a role in periodontal medicine linking periodontitis and systemic disease.

In conclusion, these results revealed the pathogenesis of periodontitis using a modified triple ligature model, allowing a higher-resolution analysis. With modified sampling methods, heterogeneity in cytokine expression was found among the three tissues and their different roles in IP was verified. While Thy-1.2⁻ fibroblasts/stromal cells in the PRT were identified as essential initiators of bone resorption, the IL-33/ST2 axis regulates tissue inflammation via IL-33 released from the gingival epithelium to control the function of tissue-resident macrophages, which could be counteracted by sST2 from Thy-1.2⁻ fibroblast/stromal cells in the PRT (Fig. 7). Although the specific mechanisms, such as downstream signaling and the responsible transcription factors, need to be further investigated and the disease model will still be modified in the future, our concepts and results provide a new understanding of the etiology of periodontitis.

## Methods
All experiments in this study were approved by the Institutional Animal Care and Use Committee/Genetically Modified Organisms Safety Committee/Research Safety Control Office of Tokyo Medical and Dental University.

### Mice
C57BL/6J WT mice were purchased from CLEA Japan (Tokyo, JP). The *Il1rl1⁻/⁻* mice were kindly provided by Shizuo Akira[60]. To generate mice carrying the *Il33*ᶠˡᵒˣ allele, the *Il33* locus and known restriction sites were used to construct the targeting vector backbone containing PGK-*neo* and DTA-positive and -negative selection cassettes. The targeting construct was linearized with *XhoI* and electroporated into 129/BL6 hybrid A9 embryonic stem (ES) cells. Potential homologous recombination in ES cells was diagnosed by PCR screening. Mice carrying the *frt*-flanked PGK-*neo* cassette were crossed to FLPe transgenic mice to remove the PGK-*neo* cassette. The detailed process of the generation of *Il33*ᐞ/ᐞ mice is described in the supplementary files (Supplementary Fig. 20, Supplementary Text 1)[61]. Both strains of transgenic mice were maintained on a C57BL/6J background. All mice were maintained in a 23–25 °C and 40–70% humidity-controlled room under specific-pathogen-free condition on a 12 h light cycle with *ad libitum* access to water and a standard laboratory chow diet (housed by strains). Mice were randomly allocated into experimental or control groups (for WT mice) and ensure equal age across genotypes. For μCT analyses, RT-qPCR (tissue samples), and histological experiments, 10–12-week-old age-matched male mice were used because these experiments have a range of 2 weeks. For flow cytometry (tissue samples) and RNA-Sequencing experiments, 10-week-old age-matched male mice were used (experimental/control animals were co-housed in flow cytometric experiments using WT mice). For any experiments using primary cells, 8–12-week-old age-matched male mice were used for harvesting the cell because the characteristics of bone marrow cells do not change much in this range, and we screened monocytes with cytokine stimulation as well.

### Construction of the modified triple ligature-induced periodontitis mouse model
To estimate the periodontitis-induced bone loss, a sterilized 6–0 blade silk ligature (Nitcho Industry, Tokyo, JP) was placed around only the maxillary left second molar according to a previously reported method[41] or all of the left molars using our modified method, and the contralateral teeth were left unligated to serve as baseline control (Supplementary Fig. 1a, Supplementary Movie. 1). According to the experiments, the mice were performed euthanasia with carbon dioxide and analyzed 1, 3, 5, 8, or 14 days after ligature placement (Supplementary Fig. 1b).

### μCT analyses
The maxillary samples were fixed in 4% paraformaldehyde for 24 h at 4 °C and then replaced with 70% ethanol until analysis. μCT scanning was performed with an inspeXio SMX-100CT scanner and inspeXio software (v7.2.2.1695; SHIMADZU, Kyoto, JP). Three-dimensional microstructural image data were reconstructed, and bone destruction was evaluated using TRI/3D-BON-FCS software (v10.01.37.47-H-64; RATOC, Osaka, JP). For evaluation, the distance between the cementoenamel junction and alveolar bone crest (the CEJ–ABC distance) was measured at six predetermined sites around the tooth for the first and

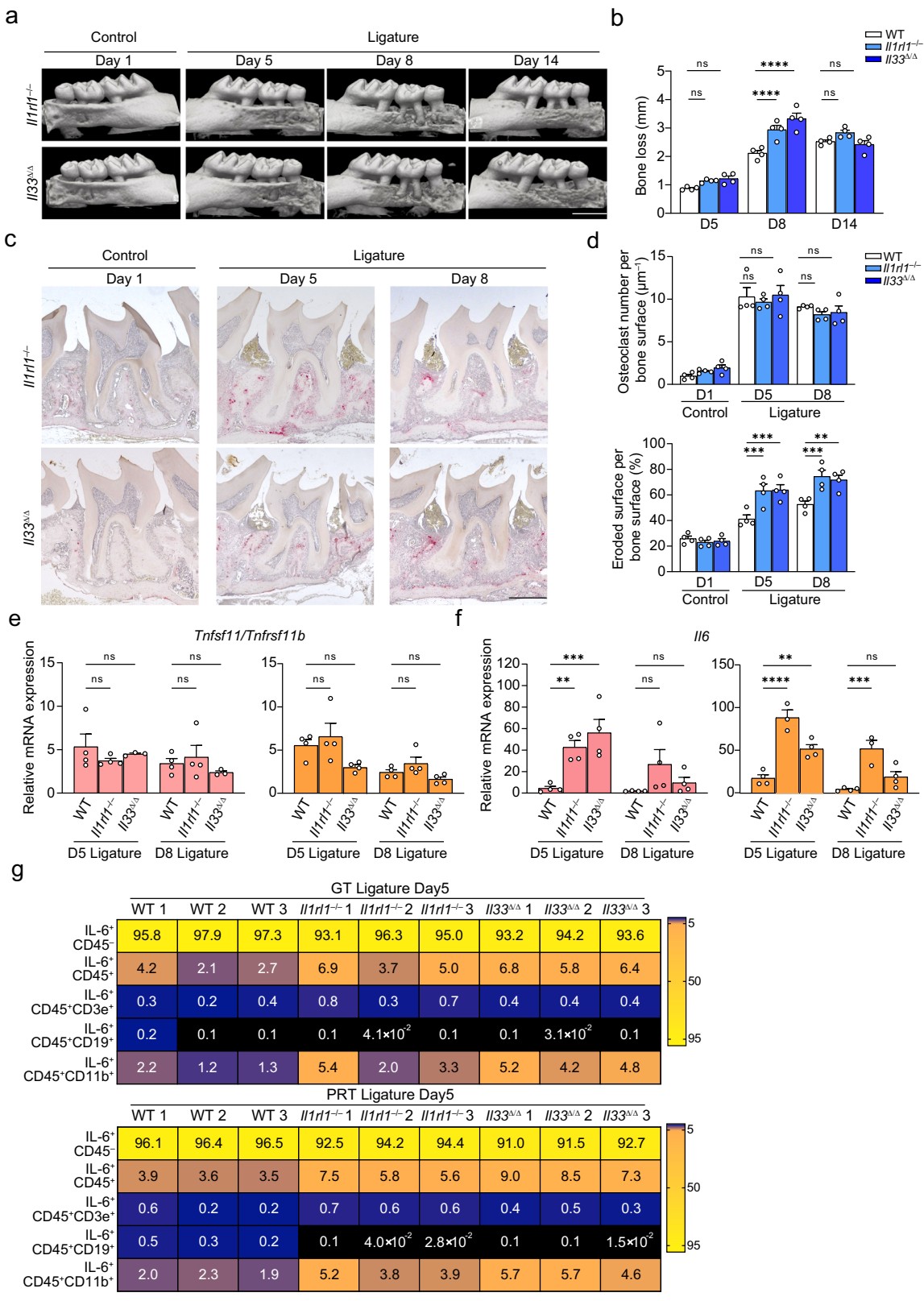

second molars and at two predetermined sites around the tooth for the third molar on the 2D interface at a fixed spatial position referenced to the center of crowns and cusps. The bone loss for each tooth was calculated by subtracting the CEJ–ABC distance between the ligature and control sides at each site and summing them. Total bone loss was calculated by summing the bone loss in three teeth.

## Histological analysis

After μCT analyses, maxillary samples were decalcified in EDT-X (Falma, Tokyo, JP) for 3 weeks and embedded in paraffin after dehydration. Four μm sections were made centered on the second molar site. Sections were stained with TRAP[61–63], and Picro-Sirius Red (ScyTek, Logan, US) respectively. The bone surface area around the second molar root was defined as the bone surface for normalization. TRAP-

**Fig. 5 | Ligature placement of knockout mice indicating that the IL-33/ST2 axis plays a protective role in acute phase. a** Representative maxillary µCT images of four independent mouse experiments on day 5, 8 and 14. Scale bar, 1 mm. **b** Total bone loss in knockout mice compared to WT mice (n = 4 mice per group). **c** Representative TRAP staining images of four independent mouse experiments on day 5 and 8. Scale bar, 300 µm. **d** Upper: Temporal change in the osteoclast number per bone surface in the second molar region (n = 4 mice per group); Lower: Temporal change in the eroded surface per bone surface in the second molar region (n = 4 mice per group). **e, f** Temporal change in the *Tnfsf11/Tnfrsf11b* ratio (**e**) and *Il6* (**f**) in the GT and PRT between mouse strains (control Day 5 of each tissue in each

strain = 1; n = 4 mice per group except for the groups highlighted in Source Data file, which were n = 3 mice per group). **g** Heat map of the positive percentage of different IL-6+ lineages in the ligated GT and PRT. The data from three independent experiments are shown (n = 3 mice per group, GT/PRT 1, 2, 3). GT gingival tissue, PRT peri-root tissue. Data are presented as the mean ± SEM. *P < 0.05; **P < 0.01; ***P < 0.001; ****P < 0.0001; ns (not significant), P > 0.05; by two-way ANOVA with multiple comparisons via Dunnett's test (**b, e, d, f**). The obvious outliers were evaluated and excluded by Grubb's test (α = 0.05). The related gating strategy is shown in Supplementary Fig. 29 (**g**). Exact P values are presented in Supplementary Data. 1. Source data are provided as a Source Data file.

positive multinucleated (> 3 nuclei) cells in the bone surface area were counted as osteoclasts, and the crenated or lacunar bone surface area was defined as the eroded surface. All images were obtained by the all-in-one fluorescence microscope BZ-X700 (KEYENCE, Osaka, JP), and the eroded surface and bone surface were measured with a BZ-X analyzer (v.1.4.0.1; KEYENCE).

### Immunofluorescence staining

After excluding circulating blood by PBS perfusion of the mice, maxillary samples were embedded immediately in SCEM-(L1) (SECTION-LAB, Yokohama, Japan) and immersed in hexane refrigerated by dry ice. Four µm thickness nonfixed and nondecalcified sections were made by Kawamoto's methods[64] and preserved in a −80 °C refrigerator after 4-h air drying in a −20 °C cryostat (Leica Biosystems, Wetzlar, Germany). Sections were rewarmed to −30 °C for 30 min before starting the stain and immediately immersed in cold 4% paraformaldehyde phosphate buffer (Nacalai Tesque, Kyoto, JP) for post-fixation (all the reaction after this step was performed at room temperature). Then, the section was washed three times using PBS and sequentially blocked by blocking one histo (Nacalai Tesque), 0.1% streptavidin (Nacalai Tesque) PBS solution, 0.01% biotin (Nacalai Tesque) 300 mM NaCl PBS solution. The antibodies were diluted in Can Get Signal immunostain solution B (TOYOBO, Osaka, Japan) with 5% v/v of blocking one histo (only polyclonal antibodies), and sequentially stained in a fixed order or only the antibodies related to cytokine staining (rat/rabbit primary antibody, goat primary antibodies, immune-HRP-polymer conjugated anti-goat secondary antibody, Alexa Fluor 594 or Alexa Fluor 647 conjugated anti-rat/rabbit secondary antibody). For endogenous HRP quenching, we treated sections with 0.1% $H_2O_2$ (Nacalai Tesque) PBS solution between the step of goat primary antibody and anti-goat secondary antibody. Every staining step was performed for one hour, protecting from light, accompanied by three times of PBS washes between every staining step. After finishing the antibody staining, the section was washed two times with pH 8.5 0.05 M borate buffer (Nacalai Tesque) and then reacted with 1 µg ml⁻¹ biotinyl tyramide (Sigma-Aldrich, St. Louis, US) buffered in 0.05 M borate buffer contains 0.003% $H_2O_2$ for 10 min. Washing sections four times using PBS and then reacting them with Alexa Fluor 488 conjugated streptavidin for 2 h, accompanied by another four times of PBS washes. Finally, the sections were treated with the vector trueview autofluorescence quenching kit (Vector laboratories, Newark, US) as the manufacturer suggested, stained with 5 µg ml⁻¹ DAPI diluted in PBS, and sealed with VECTASHIELD Vibrance Antifade Mounting Medium (Vector laboratories). All images were obtained by the all-in-one fluorescence microscope BZ-X700 (KEYENCE) after thoroughly hardening the mounting medium, with fixed imaging conditions. The nonspecific signal in cytokine stain on tooth enamel was observed in all sections. The area of the gingival region, the periodontal ligament region, and the length of the bone surface were measured with a BZ-X analyzer software (v.1.4.0.1; KEYENCE) according to demand.

Non-conjugated polyclonal antibodies for first staining of secretable proteins were purchased from R&D Systems (Minneapolis, US): anti-mouse IL-1 beta /IL-1F2 (1:500), anti-mouse IL-6 (1:200), anti-mouse IL-33 (1:250), anti-human/mouse TNF-alpha (1:100); and from

Abcam plc (Cambridge, UK): MMP9 (1:300). Non-conjugated monoclonal antibodies for first staining of membrane proteins were purchased from Biolegend (San Diego, US): CD31 (390; 1:200), CD45 (30-F11; 1:100), EpCAM (G.8.8; 1:50), and Thy-1.2 (30-H12; 1:200). Conjugated polyclonal antibodies for secondary staining were purchased from Biolegend: Alexa Fluor 594 goat anti-rat IgG (1:100) and Alexa Fluor 647 donkey anti-rabbit IgG (1:500); and from Nichirei Bioscience Inc (Tokyo, JP): HRP-labeled amino acid polymer-conjugated rabbit-anti goat Ig, which is ready-to-use.

### Primary cell culture

Bone marrow cells were collected from the femurs and tibias of mice (WT and *Il1rl1⁻/⁻*). Cells (1 × 10⁶ cells ml⁻¹; 1 ml for cell culture treated 12-well plate, CORNING, Corning, US; 2 ml for untreated 6-well plate, AGC TECHNO GLASS, Shizuoka, JP) were cultured in BMDM induction medium consisting of RPMI1640 (Nacalai Tesque) supplemented with 10% heat-inactivated fetal bovine serum (FBS; Biological Industries, Beit HaEmek, IL), 100 U ml⁻¹ penicillin, 100 µg ml⁻¹ streptomycin (Thermo fisher scientific, Waltham, US), pH 7.3 10 mM HEPES (Nacalai Tesque) and 20 ng ml⁻¹ rmM-CSF (Thermo fisher scientific) for five days with the addition of the same volume of medium on day 3. In the polarization experiments, BMDM were stimulated with 25 ng ml⁻¹ rmIL-33 (Thermo Fisher Scientific), 25 ng ml⁻¹ INF-γ (Thermo Fisher Scientific; for M1), 10 ng ml⁻¹ LPS (Thermo Fisher Scientific; for M1), 25 ng ml⁻¹ rmIL-4 (Thermo Fisher Scientific; for M2a), 50 ng ml⁻¹ rmIL-10 (R&D Systems; for M2c) or a combination of these for an additional two days, then evaluated with flowcytometry or RT-qPCR.

### RT-qPCR

Maxillary samples were fixed in RNAlater (Sigma-Aldrich) for 12 h at 4 °C and separated into the GT, PRT, and BT (Supplementary Fig. 3a; sampling of single ligature model was only performed in the tissues around the second molar) in RNAlater. Total RNA was isolated using TRI Reagent (MRC, Cincinnati, US) and a Direct-zol-96 RNA Kit (Zymo Research, Irvine, US) according to the manufacturer's instructions, separately for the three tissues. For cell samples in macrophage polarization experiments, total RNA was isolated using TRI Reagent and purified as the manufacturer suggested. First-strand cDNAs were synthesized with PCR Thermal Cycler Dice Touch (TaKaRa Bio, Shiga, JP) using ReverTra Ace qPCR RT Master Mix with gDNA Remover (TOYOBO). RT-qPCR analysis was performed with the CFX384 Touch Real-Time PCR Detection System and Bio-Rad CDX Manager software (v3.11517.0823; Bio-Rad Laboratories, Hercules, US) using THUNDERBIRD Next SYBR qPCR/UNG Set (TOYOBO). All genes were measured using the primers listed in Supplementary Table 4, 5. The relative expression levels normalized to internal reference control (*Eef2* for tissue samples; *Stx5a* for macrophage samples) were calculated using the standard curve method.

### RNA-Sequencing and data analysis

RNA samples of the three tissues on day 5 were collected as RT-qPCR. A total of four RNA samples were taken from each tissue, on both the ligature and control sides, and subjected to RNA-Seq. RNA-Seq libraries were prepared from total RNA using poly (A) enrichment of mRNA

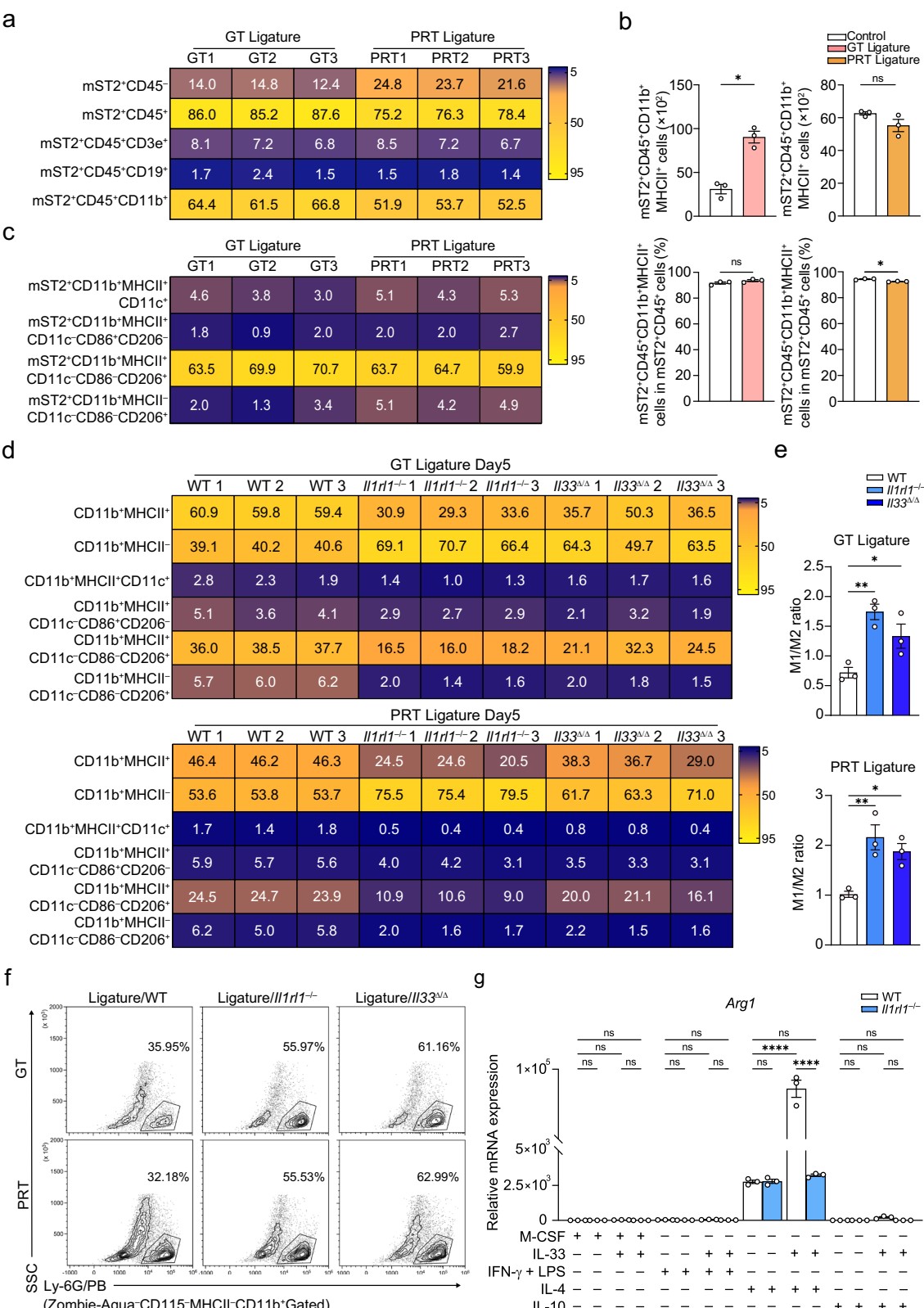

(mRNA-Seq) to remove ribosomal RNA. The adjusted libraries were sequenced with 2 × 150-bp paired-end reads on a NovaSeq 6000 system (Illumina, San Diego, US). Low-quality and adapter sequences from the paired-end reads were trimmed using Trimmomatic (v0.39)[65]. The quality of the trimmed reads was assessed using FastQC (v0.11.9). Trimmed paired-end reads were aligned to the mouse reference genome (GRCm39) using HISAT2 (v2.2.1)[66]. StringTie (v2.1.7)[67] was used to assemble the transcripts and estimate the abundance based on GRCm39 gene annotations. Differential expression analysis was conducted using the DESeq2 (v1.32.0) package for R[68] (adjusted $P$ value < 0.05; false discovery rate < 0.1; fold-change of normalized counts ≥ 2). GO/KEGG Analysis of DEGs was performed by the module based on clusterProfiler[69] for R in Hiplot (https://hiplot.cn/advance/clusterprofiler-go-kegg) with the latest version of the KEGG dataset

**Fig. 6 | mST2 positive cell populations and alteration of cell composition in mice lacking *Il33* or *Il1rl1*. a, c** Heat map of the percentage of different mST2⁺ lineages in the ligated GT and PRT with two gating strategies (**a**, in live cells; **c**, in mST2⁺CD11b⁺ cells). The data from three independent experiments are shown (*n* = 3 mice per group, GT/PRT 1, 2, 3). **b** Changes in the cell number and percentages (in mST2⁺CD45⁺ cells) of mST2⁺CD45⁺CD11b⁺MHCII⁺ lineages in the GT and PRT (*n* = 3 mice per group). **d** Heat map of the percentage of different myeloid sublineages (in CD11b⁺ cells) in the ligated GT and PRT. The data from three independent experiments are shown (*n* = 3 mice per group, GT/PRT 1, 2, 3). **e** Changes in the ratio of M1 macrophage (CD11b⁺MHCII⁺CD11c⁻CD86⁺CD206⁻ cells) number to M2 macrophage (CD11b⁺MHCII⁻CD11c⁻CD86⁻CD206⁺ cells) number in the ligated GT and PRT (*n* = 3 mice per group). **f** Changes in Ly-6G⁺ neutrophils in the ligated GT and PRT with

ligature placement. A representative contour plot of three independent mouse experiments is shown, and the percentages presented in the gates are the mean of them. **g** mRNA expression of *Arg1* of the in vitro polarized macrophages differentiated from the BMDM of WT and *Il1rl1⁻/⁻* mice. WT M0 Control (no treatment with rmIL-33) = 1; *n* = 3 mice per group. GT gingival tissue, PRT peri-root tissue, BMDM bone marrow-derived macrophage. Data are presented as the mean ± SEM. **P* < 0.05; ***P* < 0.01; ****P* < 0.001; *****P* < 0.0001; ns (not significant), *P* > 0.05; by two-side unpaired t-test with Welch's correction (**b, e**); and by two-way ANOVA with multiple comparisons via Tukey's test (**g**). The related gating strategy was shown in Supplementary Fig. 29 (**a, b**), 31 (**c–e**), 32 (**f**). Exact *P* values are presented in Supplementary Data. 1. Source data are provided as a Source Data file.

(mmu_kegg_20220421.rds) and org.Mm.eg.db annotation package provided in the website (KEGG: adjusted *P* value < 0.05; false discovery rate < 0.1; GO: adjusted *P* value < 0.01; false discovery rate < 0.05). Immune cell composition prediction was performed by the ImmuCC algorithm on the authors' server[44] (http://218.4.234.74:3200/immune/). Gene set enrichment analysis (preranked fgsea) was performed using iDEP 1.1.1 (http://bioinformatics.sdstate.edu/idep11/)[70] as developers recommended (all steps were performed with default settings).

## Flow cytometry

For flow cytometry analysis of the periodontal tissue, ligatures were placed on both sides of the maxillary molars for 5 days as the experimental group; non-treated mice were used as the control group. After euthanasia, 500 µl blood was collected from the heart and maintained in a 1.5 ml tube containing EDTA until tissue dissolution was finished. Maxillary samples were separated into the GT, PRT, and BT (Supplementary Fig. 3a) in PBS supplemented with 2% heat-inactivated FBS and maintained in DMEM (Thermo Fisher Scientific) supplemented with 10% heat-inactivated FBS, 100 U ml⁻¹ penicillin, 100 µg ml⁻¹ streptomycin (Thermo Fisher Scientific), 0.25 mg ml⁻¹ DNase I (Worthington, Lakewood, US) and pH 7.0 25 mM HEPES (Nacalai Tesque) until tissue dissolution. Two to four specimens were pooled into a single sample for further analysis (two for general analysis or analyzing tissue component cells, three for analyzing myeloid cell subsets, and four for analyzing T cells).

After crushing the BT with scissors, tissues were incubated in the digestion buffer consisting of DMEM supplemented with 6% heat-inactivated FBS, 100 U ml⁻¹ penicillin, 100 µg ml⁻¹ streptomycin, pH 7.0 25 mM HEPES (Nacalai Tesque), 0.15 mg ml⁻¹ DNase I, and 4 or 1 mg ml⁻¹ Dispase II (Roche, Basel, CH) (4 mg ml⁻¹ for analyzing tissue component cells or general cell composition, 1 mg ml⁻¹ for analyzing myeloid cell and T cell subsets) at 37 °C for 30 min with 65 rpm horizontal shaking (TAITEC, Saitama, JP). The GT was temporarily removed from the buffer and thoroughly minced after its epithelial layer was peeled off. Then the 1:1 mixture of Collagenase I and Collagenase II (Worthington) was added to a final concentration of 1 mg ml⁻¹ and incubated for another 30 min as previously. The reaction was terminated by adding EDTA (Nacalai Tesque) for a final concentration of 5 mM. Tissue fragments were centrifuged at 350 × *g*, washed with 2% FBS/PBS, and filtered with a 70 µm cell strainer (CORNING, Corning, US) to obtain a single-cell suspension. Red blood cell lysis was performed for the BT and blood samples with Hybri-Max™ (Sigma-Aldrich) according to the manufacturer's instructions. Live cells in each sample were counted, and 400,000 cells for the GT, 200,000 cells for the PRT, 1,000,000 cells for the BT, and 5,000,000 cells for the PMBC were kept for the downstream treatment.

Before staining, cells were stimulated in a 37 °C incubator with Cell Activation Cocktail and Brefeldin A (Biolegend) for 6 h. Cells were stained with a Zombie Aqua™ Fixable Viability Kit to label the dead cells, and their Fc receptor was blocked with TruStain FcX™ PLUS (Biolegend). After the surface staining with antibodies and True-Stain Monocyte Blocker™ (Biolegend), cells were fixed with FluoroFix™

Buffer (Biolegend). For the experiments except for T cell analysis, the cells were permeabilized and stained with antibodies against cytokines diluted in Intracellular Staining Permeabilization Wash Buffer (Biolegend), except for mST2. For the analysis of T cells, the cells were sequentially permeabilized with fixation diluent and permeabilization buffer in True-Nuclear Transcription Factor Buffer Set (Biolegend), and stained with antibodies against cytokines and transcription factors diluted in the permeabilization buffer. Flow cytometric analysis was performed using CytoFLEX S with CytExpert (v2.3.1.22) and Kaluza (v2.1) software (Beckman Coulter, Brea, US). Cells were re-filtered with a 35 µm cell strainer (CORNING) before loading to flow cytometry.

For flow cytometry analysis of samples in macrophage polarization experiments, cells were washed with warm PBS (Nacalai Tesque) once and detached from multiwell plates with 15-min-incubation with 10 mM EDTA/PBS in the incubator. After washing, cells were stained for surface markers same as tissue samples but not fixed, and then directly analyzed as tissue samples.

Antibodies conjugated with fluorescein isothiocyanate (FITC), Pacific Blue (PB), phycoerythrin (PE), PerCP-Cy5.5, allophycocyanin (APC), Alexa Flour 647 (AF647), PE-Cy7, and APC-Cy7 were used for cell surface staining and intracellular staining. The anti-mouse IL-33 monoclonal antibody (IC3626R; 1:100) was purchased from R&D Systems. The following antibodies were purchased from Biolegend; anti-mouse CD3ε (145-2C11; 1:200), CD4 (GK-1.5; 1:100), CD11b (M1/70; 1:200), CD11c (N418; 1:100), CD19 (6D5; 1:200), CD31 (390; 1:200), CD45 (30-F11; 1:200), CD86 (GL-1; 1:100), CD115 (AFS98; 1:100), CD206 (C068C2; 1:100), c-kit (2B8; 1:100), EpCAM (G.8.8; 1:200), F4/80 (BM8; 1:200), I-A/I-E (M5/114.15.2; PerCP-Cy5.5 1:400; FITC 1:200), IL-6 (MP5-20F3; 1:100), IL-10 (JES5-16E3; 1:100), IL-17A (TC11-18H10.1; 1:100), Ly-6G (1A8; 1:100), RANKL (IK22/5; 1:100), Sca-1 (D7; 1:200), ST2 (DIH9; 1:50 dilution for surface staining, 1:100 dilution for intracellular staining), Thy-1.2 (30-H12; 1:400). Specific information of all antibodies and gating strategies of different experiments was described in the Reporting Summary and Supplementary Fig. 29–33, respectively.

## Statistics and reproducibility

In each experiment, multiple mice were analyzed as biological replicates. All data are presented as the mean ± standard error of the mean (SEM) values of independent replicates. The sample size for each experimental condition is indicated in the figure or corresponding figure legends. Sample sizes were based on those used in similar published studies and no statistical method was used to predetermine sample size. For obvious outliers that may be related to technical error, we performed Grubb's test (*α* = 0.05) to analyze and exclude them. The investigators were not blinded to allocation during the experiments and outcome assessment. For comparisons between two groups, significance was determined by the two-tailed unpaired *t* test with Welch's correction. For comparisons among more than two groups, one-way analysis of variance (ANOVA) along with Dunnett's or Tukey's multiple-comparison test was used. For comparisons among grouped data, two-way ANOVA with Dunnett's, Šídák's, or Tukey's multiple-comparison test was used. *P* < 0.05 was considered statistically significant

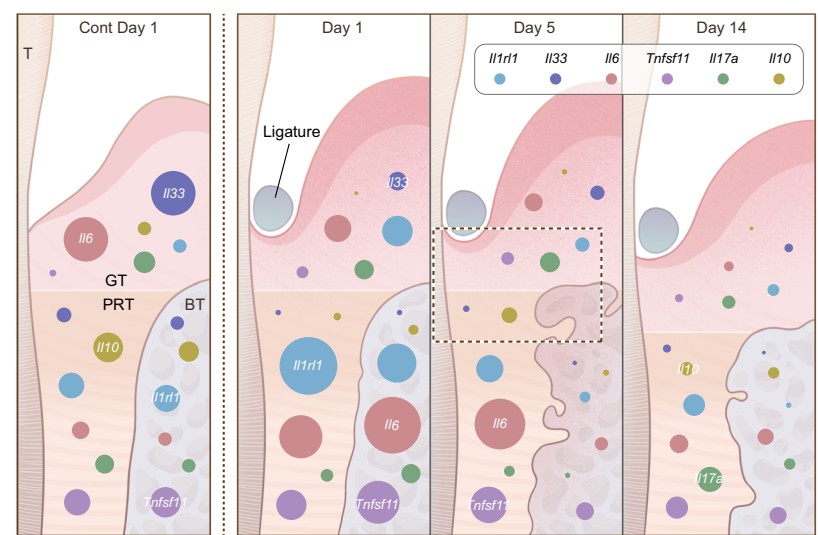

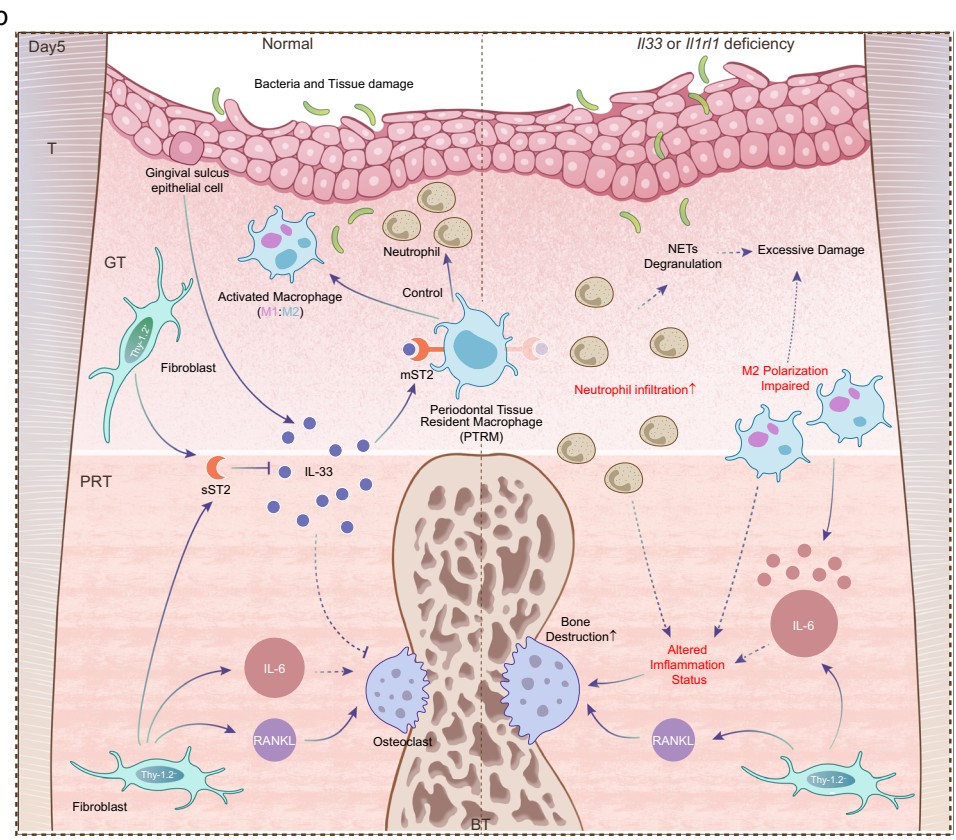

**Fig. 7 | Schematic illustration of inflammatory gene expression profiles in the course of periodontitis and the role of the IL-33/ST2 axis against acute inflammation. a** Visualization of the mRNA expression of key cytokines throughout the pathogenetic process of periodontitis in the three tissues. The scale of the cytokine circles is based on the minimum expression tissue on the control side on day 1. The size ratio of the circles in the normal part is disproportionate to the subsequent parts to show the expression differences between the tissues more clearly. **b** Visualization of the possible mechanism of the pathogenesis in IP and the possible working mechanism of the IL-33/ST2 axis. The diagram was divided by dot line into two parts: the left side represents the normal situation, and the right side represents the *Il33*- or *Il1rl1*-deficient situation. mST2 membrane form of ST2, sST2 soluble form of ST2, GT gingival tissue, PRT peri-root tissue, BT bone tissue.

(*$P < 0.05$; **$P < 0.01$; ***$P < 0.001$; ****$P < 0.0001$; ns (not significant), $P > 0.05$; the actual $P$ values are shown in the Supplementary Data. 1, 2). All statistical analysis was performed with Prism 9/10 (GraphPad Software, Boston, US).

### Reporting summary

Further information on research design is available in the Nature Portfolio Reporting Summary linked to this article.

## Data availability

The data that support the plots within this paper are available from the corresponding author upon reasonable request. The RNA-Sequence data have been deposited under GEO: GSE221720 and GSE244931. We also used the mouse transcriptome index *Mus musculus* GRCm39 (https://www.ncbi.nlm.nih.gov/datasets/genome/GCF_000001635.27/) and KEGG annotation data in Hiplot platform (https://download.hiplot-academic.com/api/file/fetch/?path=/f1b0ff00-3d9a-11ed-9ee1-85e3cd8

28dce/public/db/kegg/mmu_kegg_20220421.rds). Source data are provided with this paper.

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

## Acknowledgements

We are grateful to H. Takayanagi, M. Tsukasaki, Y. Yamashita, and R. Denda for providing valuable discussions. In addition, we thank Y. Tsuchiya for the technical teaching of classical mouse ligature model, T. Shimohira for the technical teaching of mouse tooth extraction, P. Lin for evaluating the quality of RNA samples used for RNA-Sequencing, T. Ono for breeding *Il1rl1*-deficient mice, C. Liu for kind help with the graphical summary, and Q. Liu for the useful suggestion article writing strategy. This work was supported in part by Advanced Research and Development Programs for Medical Innovation under JP20gm0810003 (T.N.) and JP22gm6110027 (M.H.) from Japan Agency for Medical Research and Development (AMED); Grant-in-Aid for Scientific Research (A), Scientific Research (B), and Challenging Research (Exploratory) from the Japan Society for the Promotion of Science (JSPS); Fusion Oriented REsearch for disruptive Science and Technology Program under JPMJFR200J (S.K.) from Japan Science and Technology Agency (JST); and grants from Takeda Science Foundation, Astellas Foundation for Research on Metabolic Disorders, Daiichi Sankyo Foundation, and Secom Science and Technology Foundation. The Ph.D. program of Anhao Liu was supported by MEXT scholarship.

## Author contributions

A.L. designed and performed all experiments, interpreted the results, and wrote the manuscript. M.H. provided vital advice on project planning and data interpretation and critically revised the manuscript. Y.O. and S.K. analyzed and interpreted the RNA-seq data. T.I. contributed to data interpretation and manuscript preparation. S.A. contributed to the generation of *Il1rl1*-deficient mice and provided valuable discussions. T.N. contributed to the generation of *Il33*-deficient mice, supervised the project, and modified the manuscript.

## Competing interests

All authors declare no competing interests.
