## [Peer Review File · Nature Communications]

The IL-33/ST2 axis is protective against acute inflammation during the course of periodontitisREVIEWER COMMENTS

Reviewer #1 (Remarks to the Author):

This is an interesting and well done study, but the discussion and conclusion are not supported by the evidence.

Major issues

1. It is understood that in IL-33 soluble receptor, which is protective, would worsen periodontitis in IL1rl1 deficient mice. However, it is not clear and is not discussed why IL33 deficient mice also had worse periodontitis. Are the authors implying that both IL-33 and its soluble receptor are protective? This is an oxymoron as the soluble receptor would neutralize IL-33.

2. Discussion of the role of IL-33 in promoting inflammation is minimized and critical papers are not mentioned-see examples below:

Trimarchi M, Lauritano D, Ronconi G, Caraffa A, Gallenga CE, Frydas I, Kritas SK, Calvisi V, Conti P. Mast Cell Cytokines in Acute and Chronic Gingival Tissue Inflammation: Role of IL-33 and IL-37. *Int J Mol Sci.* 2022 Oct 31;23(21):13242. doi: 10.3390/ijms232113242. PMID: 36362030; PMCID: PMC9654575.

Saluja R, Khan M, Church MK, Maurer M. The role of IL-33 and mast cells in allergy and inflammation. *Clin Transl Allergy.* 2015 Sep 29;5:33. doi: 10.1186/s13601-015-0076-5. PMID: 26425339; PMCID: PMC4588911.

Theoharides TC, Petra AI, Taracanova A, Panagiotidou S, Conti P. Targeting IL-33 in autoimmunity and inflammation. *J Pharmacol Exp Ther.* 2015 Jul;354(1):24-31. doi: 10.1124/jpet.114.222505. Epub 2015 Apr 23. PMID: 25906776.

Theoharides TC, Leeman SE. Effect of IL-33 on de novo synthesized mediators from human mast cells. *J Allergy Clin Immunol.* 2019 Jan;143(1):451. doi: 10.1016/j.jaci.2018.09.014. Epub 2018 Nov 1. PMID: 30390921.

Taracanova A, Tsilioni I, Conti P, Norwitz ER, Leeman SE, Theoharides TC. Substance P and IL-33 administered together stimulate a marked secretion of IL-1 β from human mast cells, inhibited by methoxyluteolin. *Proc Natl Acad Sci U S A.* 2018 Oct 2;115(40):E9381-E9390. doi: 10.1073/pnas.1810133115. Epub 2018 Sep 19. PMID: 30232261; PMCID: PMC6176605.

Papathanasiou E, Teles F, Griffin T, Arguello E, Finkelman M, Hanley J, Theoharides TC. Gingival crevicular fluid levels of interferon- γ , but not interleukin-4 or -33 or thymic stromal lymphopoietin, are increased in inflamed sites in patients with periodontal disease. *J Periodontal Res.* 2014 Feb;49(1):55-61. doi: 10.1111/jre.12078. Epub 2013 Apr 1. PMID: 23550893.

3. The authors should better explain whether IL-33 or its soluble receptor may be released from different cells in different stages of periodontal disease.

4. There should be an explanation as to why IL-33 or ST2 were not measured in crevicular fluid.

5. There should be an explanation why mast cells were not evaluated as they are a major source of cytokines including RANKL as IL-33.

6. The abstract does not cover all the results.

7. The title is confusing because the manuscript does not present spatiotemporal analysis of IL-33 or ST2 involvement, nor does the evidence clearly indicate "the unique role of the IL/33/ST2 axis."

8. The tables are very difficult to follow and need better titles and explanations.

9. There should be some discussion as to the relevance of the model use to human periodontal diseases.

10. There should be some discussion of clinical implications of the findings and whether the authors suggest the use of any IL-33 related treatment.

Reviewer #2 (Remarks to the Author):

The authors put together a significant body of experimental work, yet the data is largely descriptive and while they raise interesting questions, they do not investigate mechanisms of pathology in much depth.

As such the authors almost put together two separate investigations that could be further detailed and explored to understand aspects of periodontal pathogenesis.

One aspect is the "spatial investigation" of experimental periodontitis progression. Here the authors devise an approach to investigate events related to pathogenesis that occur in distinct tissue compartments; mucosal versus peri-root versus osseous. While their model is not unique (as they suggest)- it is an extension of the previous model by the Hajishengallis lab- the approach of dissecting the different compartments to evaluate immune responses in particular locations, is very interesting. The authors have some level of characterizations of the divergent immune responsiveness in the different areas – without much insight into how these participate in disease pathology. If the question is not to comprehensively understand spatial aspects of disease pathology- then the majority of the initial figures become a distraction towards the later concept of IL33/ST2 and can be largely removed to delve further into the mechanism of IL33- immunity. If however, the spatial -specific immune response is of interest – it should be further developed and explored as well as complemented with additional histological and spatial approaches that do not involve tissue dissociation and could be subject to tissue cross- contamination.

Second, the authors show limited -though interesting data of an immune-protective role of IL33/ST2 in periodontal pathology. This aspect is minimally developed and does not provide any mechanistic insights into the role of this axis in disease.

Reviewer #3 (Remarks to the Author):

The manuscript "Spatiotemporal analysis of periodontitis—the unique role of the IL-33/ST2 axis" describes a model to study periodontitis. The disease model used in this study, it is claimed, yields more tissue for analysis and enables analysis of the pathogenesis of periodontitis with different periodontal tissue components. Although this model shows some advantages over previous disease models, the mechanisms related to periodontitis involved in this study are not novel. Moreover, many experiments need more data to support the conclusion. The scope of this manuscript is more suitable for a clinical journal. Detailed information on the major points that should be addressed in this manuscript are listed below.

Major

1. The title of this manuscript is not suitable. Although analysis was conducted at different stages and in different tissues, "spatiotemporal analysis" is not a suitable description.
2. The authors performed qPCR and flow cytometry to show changes in proinflammatory cytokines and immune cells. However, these were not verified in vivo. More work needs to be done to support the authors' statement.
3. Since the authors used a triple ligature method as a novel model of periodontitis, more work should be performed to prove the statements. The authors said "the novel model ...and had little effect on the pathogenic mechanism (Fig. 1c, d)." The authors concluded that the numbers of

osteoclasts in the second molar region were similar between the single- and triple-ligature models, but in figure 1c, there seem to be fewer TRAP+ cells in the triple-ligature model, especially for days 3 and 5. Additional markers such as MMP9 and Cath K should also be tested to assess the osteoclast numbers. Also, a higher magnification view should be shown to make the signals clearly visible.

4. In Figure 2, the authors investigated the different roles of three tissues during the pathogenesis process. The data included in this part consists of mRNA expression of proinflammatory cytokines from different tissues. Some differences were pointed out by the authors among different tissues, but the authors didn't provide explanation or interpretation of these data. Moreover, mRNA expression alone is not adequate for the proposed conclusion.

5. The authors found angiogenesis in PRT, and state that inflammation in PRT may contribute to the process of IP. However, the proinflammatory cytokines were significantly increased in GT after ligature placement (Figure 2). Is there any angiogenesis in PRT in the coronal and lateral root regions, or is it only detected in the furcation? Combined with the increased proinflammatory cytokines in GT (Figure 2) and destruction of type 1 collagen fibers in the coronal region (supplementary Figure 4), it looks like inflammation in GT is crucial for the process of IP.

6. In supplementary Figure 4, the figure for Day 8 seems not to be from the coronal position. Some images appear to be coronal, while some are in the root regions. It would be better to keep them consistent, and only high-quality sections should be shown.

7. qPCR results alone can't support the statement that 'inflammation in PRT may contribute to the process of IP'. The expression of these proinflammatory cytokines should be stained to show their patterns in the three tissues. Further functional studies are required.

8. Several conclusions proposed by authors are solely based on gene expression or cell type composition. For example, in the result section "The initiation of a localized pathology and possibly a systemic termination", the authors propose that "osteoclastogenesis may have been promoted by the inflammation that caused tissue destruction in GT and PRT", based on cell types they found in the different tissues, without any in vitro or in vivo data to connect the cause and the results. The authors need to validate these results. Most conclusions proposed by the authors are just hypotheses.

9. The authors claimed that the IL-33/ST2 axis played a protective role in IP and AP, but didn't determine the source of IL-33 and ST2. For example, the authors identified a possible source of IL-33 in epithelial cells in GT, stromal cells in PRT, or immune cells in both tissues. This issue can be resolved with in vivo staining with different cell type markers combined with IL-33 staining.

10. In Figure 6a, *Il33* Δ/Δ and *Il1r1* $^{-/-}$ mice, severe bone loss was shown at Day 8, but this lost bone was apparently restored at Day 14. How can the authors explain this? Does this mean the bone was regenerated or are there different roles for the IL-33/ST2 axis at different stages? What are the cellular changes at Day 14?

11. The cellular sources of IL-33 and *Il1r1* as well as their distribution and changes in the three tissues should be detected in *Il33* Δ/Δ and *Il1r1* $^{-/-}$ mice to verify the importance of the IL-33/ST2 axis.

Minors

1. The authors should annotate the figures to point out the signals or tissues they want to show, for the readers who may be less familiar with periodontitis.

2. In Figure 1e, why did the RNA yield increase in the triple ligature model? How does the increased RNA yield support spatiotemporal analysis of the different tissues?

3. It would be better to use a marker like CD31 to show angiogenesis.

Responses to reviewers' comments on the Nature Communications Manuscript (NCOMMS-23-11381A)

We are grateful to the three reviewers for their positive comments and helpful suggestions for improving the submitted manuscript. In response to the issues raised by the reviewers, we have incorporated new data after performing additional experiments, and carefully revised the manuscript. We believe that the revised manuscript has taken into consideration essentially all of the comments, and hope that it has been improved to the satisfaction of the editors and reviewers. As we moved a large part of the data from the main figures to the supplementary and added figures based on the previous results, the figure number in our new manuscript has obviously altered. Changes between two version of manuscripts were described as below:

Reviewer Only Table 1(Table. R1): Figure contents in previous manuscript.

Contents in Previous manuscript	Alteration	Location in Revised manuscript
Fig. 1	Partially moved to Supplementary Fig	Fig. 1 and Supplementary Fig. 1c, 3b
Fig. 2	Combined with other Figures	Fig. 1d, e and Supplementary Fig. 4
Fig. 3	Changed figure number	Fig. 2
Fig. 4	Changed figure number	Fig. 3
Fig. 5	Partially moved to Supplementary Fig	Fig.4 and Supplementary Fig. 15/16c, d
Fig. 6	Changed figure number	Fig. 5
Fig. 7	Modified based on additional experiments	Fig. 7
Supplementary Fig. 1	Changed figure number	Supplementary Fig. 1
Supplementary Fig. 2	Changed figure number	Supplementary Fig. 3a, c
Supplementary Fig. 3	Changed figure number	Supplementary Fig. 4b, e
Supplementary Fig. 4	Modified based on additional experiments	Supplementary Fig. 6
Supplementary Fig. 5	Changed figure number	Supplementary Fig. 7
Supplementary Fig. 6	Changed figure number	Supplementary Fig. 8
Supplementary Fig. 7	Changed figure number	Supplementary Fig. 9
Supplementary Fig. 8	Changed figure number	Supplementary Fig. 10
Supplementary Fig. 9	Changed figure number	Supplementary Fig. 11
Supplementary Fig. 10	Changed figure number	Supplementary Fig. 12
Supplementary Fig. 11	Changed figure number	Supplementary Fig. 13
Supplementary Fig. 12	Changed figure number	Supplementary Fig. 14

Supplementary Fig. 13	Modified based on additional experiment	Supplementary Fig. 24
Supplementary Fig. 14	Changed figure number	Supplementary Fig. 15
Supplementary Fig. 15	Changed figure number	Supplementary Fig. 16
Supplementary Fig. 16	Changed figure number	Supplementary Fig. 20
Supplementary Fig. 17	Changed figure number	Supplementary Fig. 21
Supplementary Table. 1	Changed table number	Supplementary Table. 5
Supplementary Table. 2	Changed table number and modified	Supplementary Table. 1
Supplementary Table. 3	Changed table number and modified	Supplementary Table. 2

Reviewer Only Table 2 (Table. R2): Description of new contents.

Additional figures	Description about contents
Fig. 4g, h	Immunofluorescence staining of ST2 and IL-33
Fig. 5g	Investigation of the increased source of IL-6 in deficient mice
Fig. 6	mST2 positive cell populations and alteration of cell composition in mice lack Il33 or Il1r1
Supplementary Fig. 1c	Temporal changes in osteoclast number and bone surface in the second molar region
Supplementary Fig. 2	The confirmation of TRAP staining via the immunofluorescence staining of MMP9
Supplementary Fig. 5	Temporal changes in immunofluorescence staining of pro-inflammatory cytokines
Supplementary Fig. 6b	CD31 immunofluorescence staining in the second molar's region
Supplementary Fig. 17	Immunofluorescence staining of ST2 on Day 5 with higher resolution
Supplementary Fig. 18	Immunofluorescence staining of IL-33 on Day 5 with higher resolution
Supplementary Fig. 19	Investigating the IL-33 high productive cells and the location of IL-33 within the cell
Supplementary Fig. 22	IL-6 positive cell component alteration among the three mouse strains
Supplementary Fig. 23 and Table. 4, 5	Comprehensive analysis of BT and PRT between WT and Il33 -deficient mice
Supplementary Fig. 24	mST2 positive cell component alteration in the GT and PRT
Supplementary Fig. 25	mST2-positive myeloid cell component alteration in the GT and PRT
Supplementary Fig. 26	Antigen-presenting myeloid cell component alteration among the three mouse strains
Supplementary Fig. 27	Non-antigen-presenting myeloid cell component alteration among the three mouse strains
Supplementary Fig. 28	Cell component alteration of helper T cell lineages among three mouse strain
Supplementary Fig. 29-33	Gating strategies

To avoid confusing the reviewers, we note that all the figure numbers and line numbers in the response letter below are from the **revised manuscripts**; none belong to the previous one. Please find below our point-by-point response to each comment by the reviewers.

Reviewer #1

We would like to thank the reviewer #1 for the helpful and invaluable comments. We hope that the revisions we have made and the new data we have included are satisfactory.

1. It is understood that in IL-33 soluble receptor, which is protective, would worsen periodontitis in *Il1rl1* deficient mice. However, it is not clear and is not discussed why *Il33* deficient mice also had worse periodontitis. Are the authors implying that both IL-33 and its soluble receptor are protective? This is an oxymoron as the soluble receptor would neutralize IL-33.

We apologize that we did not make this clear in the previous version. The relationship between the gene, transcript variants, and the protein isoform of ST2 was really complicated. The soluble (sST2) and membrane (mST2) forms of ST2 are translated by different transcript variants encoded by the same gene, *Il1rl1* (NCBI, <https://www.ncbi.nlm.nih.gov/gene/17082>; protein isoform a = mST2, isoform b = sST2 in our manuscript; the number of transcript variants are same in our manuscript). Mice deficient in this gene lack the expression of both isoforms of ST2 and lose both the ability to transduce IL-33/ST2 signaling and to competitively inhibit IL-33 binding to its receptor. To make sure which effect is dominant in periodontitis, we tested *Il33*-deficient mice at the same time. If the phenotype of *Il1rl1*-deficient and *Il33*-deficient mice is similar, it would mean that the phenotype is induced by the loss of the IL-33/ST2 axis and not by the other possibilities.

Based on this premise, our results showed consistency in the exacerbation of inflammation in periodontitis in both strains of deficient mice (Fig. 5), suggesting that this axis has a protective effect through the signal transduction. In the revised version of the manuscript, we have improved the description of the unique encoding pattern of *Il1rl1* in the introduction (lines 76-79).

2. Discussion of the role of IL-33 in promoting inflammation is minimized and critical papers are not mentioned-see examples below:

Trimarchi M, Lauritano D, Ronconi G, Caraffa A, Gallenga CE, Frydas I, Kritas SK, Calvisi V, Conti P. Mast Cell Cytokines in Acute and Chronic Gingival Tissue Inflammation: Role of IL-33 and IL-37. *Int J Mol Sci.* 2022 Oct 31;23(21):13242. doi: 10.3390/ijms232113242. PMID: 36362030; PMCID: PMC9654575.

Saluja R, Khan M, Church MK, Maurer M. The role of IL-33 and mast cells in

allergy and inflammation. *Clin Transl Allergy*. 2015 Sep 29;5:33. doi: 10.1186/s13601-015-0076-5. PMID: 26425339; PMCID: PMC4588911.

Theoharides TC, Petra AI, Taracanova A, Panagiotidou S, Conti P. Targeting IL-33 in autoimmunity and inflammation. *J Pharmacol Exp Ther*. 2015 Jul;354(1):24-31. doi: 10.1124/jpet.114.222505. Epub 2015 Apr 23. PMID: 25906776.

Theoharides TC, Leeman SE. Effect of IL-33 on de novo synthesized mediators from human mast cells. *J Allergy Clin Immunol*. 2019 Jan;143(1):451. doi: 10.1016/j.jaci.2018.09.014. Epub 2018 Nov 1. PMID: 30390921.

Taracanova A, Tsilioni I, Conti P, Norwitz ER, Leeman SE, Theoharides TC. Substance P and IL-33 administered together stimulate a marked secretion of IL-1 β from human mast cells, inhibited by methoxyluteolin. *Proc Natl Acad Sci U S A*. 2018 Oct 2;115(40):E9381-E9390. doi: 10.1073/pnas.1810133115. Epub 2018 Sep 19. PMID: 30232261; PMCID: PMC6176605.

Papathanasiou E, Teles F, Griffin T, Arguello E, Finkelman M, Hanley J, Theoharides TC. Gingival crevicular fluid levels of interferon- γ , but not interleukin-4 or -33 or thymic stromal lymphopoietin, are increased in inflamed sites in patients with periodontal disease. *J Periodontal Res*. 2014 Feb;49(1):55-61. doi: 10.1111/jre.12078. Epub 2013 Apr 1. PMID: 23550893.

We appreciate the suggestion of reviewer #1 and agree with it that we should increase some paragraphs and references about the inflammation-promoting effect of IL-33. We rewrote and mentioned the above papers in introduction (PMID: 36362030, PMID: 26425339, PMID: 30232261, PMID: 25906776, lines 79-84 and 87-89) and discussion (PMID: 23550893, lines 491-494) in the revised manuscript.

3. The authors should better explain whether IL-33 or its soluble receptor may be released from different cells in different stages of periodontal disease.

We are grateful to the reviewer #1 for this excellent suggestion. In addition to the major source of both proteins in IP previously shown by the flow cytometry and qPCR (IL-33: EpCAM⁺ in GT and Thy-1.2⁻ fibroblasts/stromal cells in PRT; sST2: Thy-1.2⁺ fibroblasts/stromal cells in GT and Thy-1.2⁻ fibroblasts/stromal cells in PRT; Fig. 3c-h, 4a-d and Supplementary Fig. 14a, b, d-g, 15, 16), we performed additional immunofluorescence staining for the intact and inflammatory periodontal tissues to better address them in our revised manuscript. In belief, we described them as follows:

IL-33

When the periodontal tissue was intact, IL-33 positive cells were found in neither PRT nor GT of the second molar region, especially no signal in EpCAM⁺ gingival sulcular epithelium cells when the periodontal tissue was intact (Fig. 4h and Supplementary Fig.18). We attempted to solve this discrepancy by searching for IL-33-positive cells in gingival tissue around second molar and found IL-33^{high} cells in the submucosal layer of gingiva located at the mesial of the first molar, where was far from the region of the tooth (Supplementary Fig.19a). These “IL-33-enriched submucosal stromal cells (IL-33SCs)” were CD45⁻ and tended to be Thy-1.2⁺ where they approached tooth, showed high IL-33 signal on and around their nucleus, and interestingly had an EpCAM⁻ gingival epithelium as a superstratum (Supplementary Fig.19a, b). We then examined the expression of IL-33 in periodontal tissues after induction of periodontitis. IL-33⁺EpCAM⁺ gingival sulcular epithelial cells (IL-33ECs) were observed in GT, while a reduction in IL-33-positive CD45⁻CD31⁻ cells (IL-33FCs) was observed in PRT (Fig. 4h and Supplementary Fig.18). The intracellular location of IL-33 in IL-33ECs was around the nucleus, as in IL-33SCs, but in IL-33FCs it was clearly outside the nucleus. Results above were described in lines 286-301, and the role of three different sources of IL-33 were discussed in lines 467–496 of the revised manuscript.

sST2

There are few ST2-positive cells in the control sample and low expression of the transcripts variant 2/3 (encoding sST2) according to the flow cytometry which we presented previously (Fig. 4a and Supplementary Fig. 15), and the qPCR results we presented previously showed few cells producing sST2 when the tissue was intact (Fig. 3c–f and Supplementary Fig.14a, b). Our immunofluorescence staining results for ST2 in the revised manuscript (Fig. 4g and Supplementary Fig. 17) show that the expression was observed in 1) some CD45⁺ immune cells and 2) few of CD45⁻EpCAM⁻Thy-1.2⁻ fibroblast/stromal cells in PRT of control samples. For the specific type of population 1, we found the mST2⁺ periodontal tissue-resident macrophages (PTRMs) in additional experiments, and this population was stably present in PRT regardless of whether inflammation was induced (Fig. 6a–c, and Supplementary Fig. 24, 25). Considering these data, population 1 consists mainly of the mST2-positive PTRMs, whereas sST2-producing cells were very few and hard to find.

When periodontitis was induced, there were three populations in the periodontal tissues: 1) CD45⁺ cells in GT/PRT; 2) CD45⁻EpCAM⁻Thy-1.2⁺ fibroblast/stromal cells in GT; 3) CD45⁻EpCAM⁻Thy-1.2⁻ fibroblast/stromal cells in PRT (Fig. 4g and Supplementary

Fig.17). Given their low number compared to CD45⁻ cells when ST2-positive cells were detected in flow cytometry (Fig. 4a and Supplementary Fig. 15), the population 1 was mainly the mST2⁺ PTRMs, following a similar rationale as the control samples (Fig.6a–c and Supplementary Fig. 24, 25). In contrast, the increased populations 2 and 3 were both sources of sST2 according to the flow cytometry of ST2/mST2 (Fig. 4a, b, 6a–c and Supplementary Fig. 15, 24, 25), as well as mild fold changes in the expressions of transcript variants encoding mST2 (Fig. 3c–f and Supplementary Fig. 14a, b). Thus, although few cells produce sST2 in normal condition, CD45⁻EpCAM⁻Thy-1.2⁺ fibroblast/stromal cells in GT and CD45⁻EpCAM⁻Thy-1.2⁻ fibroblast/stromal in PRT would express large amounts of sST2 upon induction of periodontitis, especially the latter. These results were also described in the lines 270-277, 282-301 and 334-356 in revised manuscript.

4. There should an explanation as to why IL-33 or ST2 were not measured in crevicular fluid.

As mentioned by reviewer #1, gingival crevicular fluid is a clinical examination method that is collected with paper strips and is used to help the diagnosis and prognostic assessment of periodontitis. Although there is an established method to collect the crevicular fluid from mice (S. Matsuda *et al.*, *J Immunol Methods*, 2017), the removal of the pre-placed ligature and the replacement of the sample-collecting ligature will affect the morphology of the periodontal tissue, making other analyses impossible. In addition, the concept of this study was to monitor the pathogenesis of periodontitis with higher resolution by separating periodontal tissues into GT, PRT and BT, and this approach allowed us to discover the hidden role of the IL-33/ST2 axis. Since gingival crevicular fluid is mainly produced from GT (especially gingival sulcular epithelium) and similar percolates are not produced from the other two tissues despite we examined the methods to collect them as well. We considered performing the experiments suggested by reviewer #1 in the early stages of the study, but decided against it for the reasons stated above.

5. There should be an explanation why mast cells were not evaluated as they are a major source of cytokines including RANKL as IL-33.

We have shown that the primary source of IL-33 and RANKL was CD45⁻ tissue component cells, and few were produced by CD45⁺ immune cells such as T cells (Fig. 4b, 2d–f and Supplementary Fig. 4b, e, 10a, b, 16; lines 268-270 and 184-187). Therefore,

we had not previously mentioned mast cells. However, since mast cells are important for periodontitis, as suggested by reviewer #1, we evaluated mast cells from WT and both strains of deficient mice in additional experiments. Mast cells are essential for the IL-33/ST2 axis because of their important role in allergy and respiratory disease, but they tend to be effector cells and express mST2 to receive IL-33 signals (Liew FY *et al.*, *Nat Rev Immunol*, 2016). Although there was a population of mast cells (CD11b⁺MHCII⁻Ly6G⁻c-kit⁺ cells) in our additional experiments and they enriched in mST2 expression, the major mST2-positive cells were PTRMs (Fig. 6a–c and Supplementary Fig. 24c, d, 25a–c; lines 334-356). Moreover, loss of either *Il33* or *Il1r1* significantly suppressed the differentiation or activation of mast cells, indicating that their role in the pathogenesis of periodontitis in IP is not crucial, as inflammation was exacerbated (Fig.5b and Supplementary Fig27a, b; lines 450-454). However, it is possible that mast cell-derived IL-33 also plays a hidden role in the more acute or chronic phases of periodontitis, which could be investigated further.

6. The abstract does not cover all the results.

In the revised manuscript, we have tried our best to rewrite the abstract to cover all the results after integrating the content of the additional experiments (lines 19-35).

7. The title is confusing because the manuscript does not present spatiotemporal analysis of IL-33 or ST2 involvement, nor does the evidence clearly indicates "the unique role of the IL/33/ST2 axis."

We changed the title to “IL-33/ST2 axis avoid the excessive inflammation in the pathogenesis of periodontitis” after incorporating the content of the additional experiments. We believe that the new title summarizes our study more clearly and accurately.

8. The table are very difficult to follow and need better titles and explanations.

Supplementary Table.1, 2 (number changed) were originally intended to present the P values for the bar graphs according to the journal's recommendation, so we have not yet considered their readability. We thank reviewer #1 for the suggestions and have made them easier to follow.

9. There should be some discussion as to the relevance of the model use to human periodontal diseases.

Basic research should be done for human applications. We thank the reviewer #1 for pointing out that we were missing the relevant discussions to embody this idea. We have added the discussions on the application of the model and concepts in lines 408–410, and the potential of IL-33 and ST2 in periodontal medicine in lines 498–508.

10. There should be some discussion of clinical implications of the findings and whether the authors suggest the use of any IL-33 related treatment.

Please see point 9 above.

Reviewer #2

We are very grateful to the reviewer #2 for the positive comments and constructive suggestions. We hope that the manuscript has been improved to the satisfaction of the reviewer.

The authors put together a significant body of experimental work, yet the data is largely descriptive and while they raise interesting questions, they do not investigate mechanisms of pathology in much depth.

As such the authors almost put together two separate investigations that could be further detailed and explored to understand aspects of periodontal pathogenesis.

One aspect is the “spatial investigation” of experimental periodontitis progression. Here the authors devise an approach to investigate events related to pathogenesis that occur in distinct tissue compartments; mucosal versus peri-root versus osseous. While their model is not unique (as they suggest)- it is an extension of the previous model by the Hajishengallis lab- the approach of dissecting the different compartments to evaluate immune responses in particular locations, is very interesting. The authors have some level of characterizations of the divergent immune responsiveness in the different areas – without much insight into how these participate in disease pathology. If the question is not to comprehensively understand spatial aspects of disease pathology- then the majority of the initial figures become a distraction towards the later concept of IL33/ST2 and can be largely removed to delve further into the mechanism of IL33- immunity. If however, the spatial -specific immune response is of interest – it should be further developed and explored as well as complemented with additional histological and spatial approaches that do not involve tissue dissociation and could be subject to tissue cross-contamination.

We appreciate reviewer #2's vital perspective on the issues in our previous manuscripts. As suggested by reviewer #2, our focus was not clear and the “spatial investigation” part prevented us from better presenting the function of the IL-33/ST2 axis. Therefore, we moved half of the “spatial investigation” part to the supplementary figures and added a paragraph on the IL-33/ST2 axis to explain it more clearly (Table. R1, 2). We also changed the title of our manuscript to “IL-33/ST2 axis avoid the excessive inflammation in the pathogenesis of periodontitis” for this purpose. However, we believe that the cytokine analysis and “spatial investigation” part are the basis for the reader to notice the pathogenesis different from the previous theory and the relative position of the IL-33/ST2

axis in this context. Therefore, we have moved the figures, but left the descriptions in revised manuscript. Although the main focus is no longer on spatial exploration, we have also added the immunofluorescence staining (Fig. 4g, h and Supplementary Fig. 5, 17-19) for cytokines and cell markers used in flow cytometry analysis to increase the reliability of the results. In addition, we removed imprecise terms such as “spatiotemporal” and “novel model” to improve the accuracy of our manuscripts.

Second, the authors show limited -though interesting data of an immune-protective role of IL33/ST2 in periodontal pathology. This aspect is minimally developed and does not provide any mechanistic insights into the role of this axis in disease.

Thank for the suggestion of reviewer #2 for improving our manuscript. We agree with reviewer #2 that we lack the mechanistic insight into the function of the IL-33/ST2 axis. Therefore, we performed additional experiments using flow cytometry to explore this at the cellular level. We confirmed that the increased expression of IL-6 in either IL-33- or ST2-deficient mice is derived from myeloid lineage cells and that the major mST2-rich cells in periodontal tissue are a subpopulation of a tissue-resident macrophages, which we defined as periodontal tissue-resident macrophages (PTRM). Deficiency of IL-33/ST2 signaling resulted in a shift in M1 macrophage polarization due to a decrease in M2 macrophages in both strains of deficient mice. Meanwhile, the increase in monocytes in *Il1r1*-deficient mice indicated the impacts on macrophage formation. The abnormality in macrophage function led to a more than twofold over-infiltration of neutrophils, which was also responsible for the increased IL-6 expression with PTRMs. The details of these results are shown in Fig. 5g, 6, 7 and Supplementary Fig. 22, 24–28. We hope that these results described in lines 320-385 will satisfy reviewer #2.

Reviewer #3

We are very grateful to reviewer #3 for the helpful comments and constructive suggestions that helped us in the evaluation of our model. We hope that the manuscript has been improved to the satisfaction of the reviewer.

Major

1. The title of this manuscript is not suitable. Although analysis was conducted at different stages and in different tissues, “spatiotemporal analysis” is not a suitable description.

We agree with the reviewer #3 that our previous title was misleading to readers, and inaccurate for our results. We changed the title to “IL-33/ST2 axis avoid the excessive inflammation in the pathogenesis of periodontitis.” after incorporating the content of the additional experiments.

2. The authors performed qPCR and flow cytometry to show changes in proinflammatory cytokines and immune cells. However, these were not verified in vivo. More work needs to be done to support the authors’ statement.

Thank for the constructive suggestions of reviewer #2. Although we carefully separated the tissue samples and confirmed the contamination at the nucleic acid level (Supplementary Fig. 3c), the contamination cannot be eliminated and is difficult to evaluate at the cellular level, as pointed out by reviewer #3. Therefore, to increase the reliability of our qPCR and flow cytometry results, we performed the immunofluorescence staining by combining Kawamoto’s methods (Kawamoto T and Kawamoto K, *Skeletal Development and Repair: Methods and Protocols*, 2021) and a tyramide amplification system (Stack EC *et al.*, *Methods*, 2014) to observe the cytokine expression *in vivo*. The former approach allowed us to prepare the unfixed and undecalcified samples to maximize antigen protection; the latter procedure allowed us to amplify the weak signals from cytokines despite causing a non-specific signal in the tooth enamel. The results of representative proinflammatory cytokines (IL-6, IL-1 β , TNF- α) showed a similar trends to the qPCR results in both expression level and spatial location (Supplementary Fig. 5; lines 155-157), with IL-6 expression being predominant, especially in PRT at day 5. Furthermore, we also evaluated the source of IL-33 and ST2 in the same way, which described in major point 9.

3. Since the authors used a triple ligature method as a novel model of periodontitis, more work should be performed to prove the statements. The authors said “the novel model ...and had little effect on the pathogenic mechanism (Fig. 1c, d).” The authors concluded that the numbers of osteoclasts in the second molar region were similar between the single- and triple-ligature models, but in figure1. c, there seem to be fewer TRAP+ cells in the triple-ligature model, especially for days 3 and 5. Additional markers such as MMP9 and Cath K should also be tested to assess the osteoclast numbers. Also, a higher magnification view should be shown to make the signals clearly visible.

We apologize to reviewer #3 for being skeptical of our results due to our inadequate explanation of the static evaluation of osteoclasts. It is well known that the slides of mouse teeth are difficult to prepare, especially in confined areas such as the region between the mesial-buccal root and the distal-buccal root, which makes the statics using slides less stable. Abe et al. (J Immunol Methods, 2013) also statically evaluated their model only by μ CT and whole tissue staining. We have shown that our model is similar to the previous model in this way (Fig. 1a, b). However, we considered histologic evaluation essential, so we worked on measurement and static methods to stabilize the data. After trying many methods, we decided to normalize the osteoclast number by the length of the bone surface length from the mesial top of the alveolar bone ridge to the distal one. The osteoclast number was also counted based on the area we measured, restricted to the bone surface. The raw data of Day 3 and 5 are shown below:

Reviewer Only Table 3 (Table. R3): Osteoclast number, bone surface length, and osteoclast number per bone surface length on Day3 and 5.

	Single			Triple		
	Osteoclast Number	Bone surface (μ m)	Osteoclast number per bone surface (mm^{-1})	Osteoclast Number	Bone surface (μ m)	Osteoclast number per bone surface (mm^{-1})
Day3	20	4300	4.651163	21	3526	5.955757
	27	4315	6.257242	17	3562	4.7726
	23	4454	5.163898	14	3536	3.959276
	28	4621	6.059295	20	4007	4.991265
Day5	35	3302	10.59964	37	2737	13.51845
	28	3508	7.981756	26	2804	9.272468
	40	3428	11.66861	30	3301	9.088155
	34	3707	9.171837	28	3030	9.240924

As reviewer #3 pointed out, the osteoclast number was lower in the triple ligature model than in the single ligature model at these timepoints. However, the bone surface was also shorter on Day 3, making the value of osteoclast number per bone surface (Oc. N/B. S) come to a similar level. Oc. N/B. S on Day 5 was also at a similar level as on Day 3, as Oc. N and B.S were clearly in a decreasing trend. This phenomenon may be mainly due to the fact that we prepared the slides by model type rather than by time. Considering that the preparation takes a long time and is difficult, we prepared slides of 8 blocks per day, and the slide of day 3 and 5 of the triple model came from the same working day with high probability (4 blocks/time point), which could affect the direction and angle of cutting. We added the figures of osteoclast number and bone surface length in Supplementary Fig. 1c for the reader, and removed the description "had little effect on the pathogenic mechanism" in the manuscript. If reviewer #3 considers that we should show a similar image to avoid misleading the reader, we would be happy to change the images.

Meanwhile, as reviewer #3 suggested, we also attempted to evaluate osteoclasts using immunofluorescence staining for MMP9, because recent studies have shown that CtsK in periodontal tissue is no longer osteoclast-specific (Tsukasaki M et al., Nat Commun, 2022; Kondo T et al.; Commun Biol, 2022). We restricted to counting on the bone surface, and TRAP staining in a triple ligature model showed a similar trend (Supplementary Fig. 2). However, TRAP staining seems to be better because MMP9 staining provides non-osteoclastic signals from fibroblasts. MMP9 and CtsK might work well in healthy bone (CtsK is not specific even in bone), but are barely satisfactory for osteoclast counting in complicated and pathological tissue. However, we thank reviewer #3 for the opportunity to evaluate MMP9 in this study, which indicated the participation of MMPs in the pathogenesis of periodontitis directly (lines 128-131). We plan to conduct further studies to better define its role.

For higher magnification of the images, we have provided the MMP9 staining in the revised manuscript (Supplementary Fig. 2). We found that the improper compression for the TRAP staining images makes it unclear in the previous manuscript. We can also provide higher magnification images if the original images are insufficient.

4. In Figure 2, the authors investigated the different roles of three tissues during the pathogenesis process. The data included in this part consists of mRNA expression of proinflammatory cytokines from different tissues. Some differences were pointed

out by the authors among different tissues, but the authors didn't provide explanation or interpretation of these data. Moreover, mRNA expression alone is not adequate for the proposed conclusion.

We appreciate the suggestion of reviewer #3. Please see point 2 above for the details on immunofluorescence staining obtained in additional experiments for increasing our credibility of qPCR data. Furthermore, we did not describe or discuss these results in the text in this manuscript because we focused on the protective function of the IL-33/ST2 axis in the pathogenesis, and wanted to keep the readers' focus more on them.

5. The authors found angiogenesis in PRT, and state that inflammation in PRT may contribute to the process of IP. However, the proinflammatory cytokines were significantly increased in GT after ligature placement (Figure2). Is there any angiogenesis in PRT in the coronal and lateral root regions, or is it only detected in the furcation? Combined with the increased proinflammatory cytokines in GT (Figure2) and destruction of type 1 collagen fibers in the coronal region (supplementary Figure 4), it looks like inflammation in GT is crucial for the process of IP.

Thank for the thought-provoking question from reviewer #3. We only showed the furcation area in the previous manuscript because we considered this region purely to consist of periodontal ligament tissue. But as reviewer #3 pointed out, the angiogenesis of other area also should be evaluated. In the revised manuscript, we carefully evaluated angiogenesis in different area by immunofluorescence staining for CD31 according to the suggestion based on the minor point 3 of reviewer #3 (Supplementary Fig. 6c, d). We showed an increase in CD31⁺ cells in periodontal tissues and found a clear trend toward the peak of angiogenesis at Day 5. When angiogenesis was analyzed separately in the gingival and periodontal ligament areas, both showed similar values on Day 1 or Day 3. However, only angiogenesis in the periodontal ligament showed a trend similar to that of bone destruction, increasing until Day 5 after ligature placement and decreasing thereafter. In the revised manuscript, now it was clearly shown using static images of the entire tooth and the area segmented based on the position of the alveolar ridge.

We understand that periodontitis begins with the inflammation of gingival tissue (gingivitis). The main difference between gingivitis and periodontitis is whether bone is destroyed. Since gingivitis and periodontitis are relevant but different diseases in dentistry,

we can consider Day 1/3 as the gingivitis phase and Day 8/14 as the periodontitis phase, and in our model, Day 5 is the exact time point bridging the two phases (the same in the conventional model, Fig. 1a, b). Therefore, the tissue that was significantly altered on Day 5 contributes more to bone destruction, which would have been crucial for the pathogenesis of “periodontitis” because bone destruction occurred shortly after this. As reviewer #3 pointed out, inflammation and collagen degeneration of the gingiva are important for the disease (Fig. 1e and Supplementary Fig. 4d, 6a, b). However, its value may be shifting the stage from the gingivitis phase to IP, because the mRNA expression of *Il6* and *Tnfrsf11* was significantly changed in PRT on Day 5, but not in GT on Day 1 or 3 (Fig. 1e), as well as histological changes. This suggests that PRT plays a vital role in inducing “bone destruction” and periodontitis. This does not mean that GT is unnecessary or unimportant, because all changes in PRT are based on/induced by the inflammation in GT. GT is a barrier that protects PRT and BT by keeping the inflammation within it, and its defect will cause bacterial invasion or immune cell infiltration to PRT, making bone destruction start. Thus, we considered that GT is important in inducing the changes in PRT, but PRT is crucial for bone destruction. We have included these points in the revised manuscript (lines 96-99, 160-163, and 431–434).

6. In supplementary Figure 4, the figure for Day 8 seems not to be from the coronal position. Some images appear to be coronal, while some are in the root regions. It would be better to keep them consistent, and only high-quality sections should be shown.

We apologize for the unclear presentation of the Picro-Sirius red staining images. For the Picro-Sirius red staining of collagen fibers, we have added new data that include the low magnification images of the whole second molar and images taken at higher magnification focused on the coronal and furcation positions without removing the ligature (Supplementary Fig. 6a, b). It can be seen that the collagen fiber in GT was degraded in the early stage after the ligature placement, and the bone ridge changed with the bone destruction, which made our previous images appear not to be in the same position. In the revised manuscript, we presented higher magnification images based on the lower magnification images to help the reader to recognize that the higher magnification images were obtained from the similar location. We also measured the change of collagen in the periodontal ligament region and found that the same trend as discussed in the previous manuscript.

7. qPCR results alone can't support the statement that 'inflammation in PRT may contribute to the process of IP'. The expression of these proinflammatory cytokines should be stained to show their patterns in the three tissues. Further functional studies are required.

We thank reviewer #3 for the valuable suggestion. With regard to the immunofluorescence staining of inflammatory cytokines, please refer to points 2 and 4 above. As for the functional studies, we appreciate the suggestion of reviewer #3, but found it difficult to perform them due to technical difficulties.

In designing functional experiments for the current study, there are two options: 1) using the the Cre-LoxP system to selectively block the expression of pro-inflammatory cytokines in the periodontal cells; 2) injecting AAV or antagonist locally into PRT to prevent the function of pro-inflammatory cytokines expressed in PRT.

Option 1 requires the periodontal tissue-specific promoter-driven Cre mice to ensure the tissue specificity of Cre-LoxP recombination. Only *Aspn-Cre* is known to be relatively specific for periodontal ligaments (Iwayama T et al., Development, 2022). However, according to their report, *Aspn* is only expressed in a subset of cells in PRT, and we have not confirmed that these overlap with the cytokine-producing population in our study. The *in vitro* functional experiment also requires the specific cell types in PRT responsible for the inflammation; this can only be ensured once the diversity of the population is clarified. Option 2 is also limited by the difficulty of accurate injections. The currently available commercial superfine needle is 37 gauge and its diameter was approximately 80 μm . The space between the bone and the root in mice is about 50–70 μm under physiological conditions, and even a 37-gauge needle could not be inserted into this space without breaking bone tissue. For this reason, injection was only possible for GT and not specifically for PRT.

Therefore, the functional experiments only could be performed based on subsequent studies that we need to investigate further, which should be a separate paper. We hope that reviewer #3 will understand this situation.

8. Several conclusions proposed by authors are solely based on gene expression or cell type composition. For example, in the result section "The initiation of a localized pathology and possibly a systemic termination", the authors propose that "osteoclastogenesis may have been promoted by the inflammation that caused tissue destruction in GT and PRT", based on cell types they found in the different tissues, without any in vitro or in vivo data to connect the cause and the results. The authors

need to validate these results. Most conclusions proposed by the authors are just hypotheses.

Thank for the suggestion of reviewer #3 about the accuracy of our manuscripts. We agree with reviewer #3 that there were some inappropriate expressions in the results section of the previous manuscript. In response to reviewer #3, we have rewritten the result parts that reviewer #3 mentioned, leaving only speculative expressions in the discussion to avoid them as much as possible in the result, which should be simply described (lines 165, 172-176, 198-202, and 208-211).

9. The authors claimed that the IL-33/ST2 axis played a protective role in IP and AP, but didn't determine the source of IL-33 and ST2. For example, the authors identified a possible source of IL-33 in epithelial cells in GT, stromal cells in PRT, or immune cells in both tissues. This issue can be resolved with in vivo staining with different cell type markers combined with IL-33 staining.

We appreciate the reviewer #3 for this valuable suggestion that helped us improve our research significantly. In addition to the major source of both proteins in IP previously shown by the flow cytometry and qPCR (IL-33: EpCAM⁺ in GT and Thy-1.2⁻ fibroblasts/stromal cells in PRT; sST2: Thy-1.2⁺ fibroblasts/stromal cells in GT and Thy-1.2⁻ fibroblasts/stromal cells in PRT; Fig. 3c–h, 4a–d and Supplementary Fig. 14a, b, d–g, 15, 16), we performed additional immunofluorescence staining for the intact and inflammatory periodontal tissues to better address them in our revised manuscript. In belief, we described them as follows:

IL-33

When the periodontal tissue was intact, IL-33 positive cells were found in neither PRT nor GT of the second molar region, especially no signal in EpCAM⁺ gingival sulcular epithelium cells when the periodontal tissue was intact (Fig. 4h and Supplementary Fig. 18). We attempted to solve this discrepancy by searching for IL-33-positive cells in gingival tissue around second molar and found IL-33^{high} cells in the submucosal layer of gingiva located at the mesial of the first molar, where was far from the region of the tooth (Supplementary Fig. 19a). These “IL-33-enriched submucosal stromal cells (IL-33SCs)” were CD45⁻ and tended to be Thy-1.2⁺ where they approached tooth, showed high IL-33 signal on and around their nucleus, and interestingly had an EpCAM⁻ gingival epithelium as a superstratum (Supplementary Fig. 19a, b). We then examined the expression of IL-

33 in periodontal tissues after induction of periodontitis. IL-33⁺EpCAM⁺ gingival sulcular epithelial cells (IL-33ECs) were observed in GT, while a reduction in IL-33-positive CD45⁻CD31⁻ cells (IL-33FCs) was observed in PRT (Fig. 4h and Supplementary Fig. 18). The intracellular location of IL-33 in IL-33ECs was around the nucleus, as in IL-33SCs, but in IL-33FCs it was clearly outside the nucleus. Results above were described in lines 286-301, and the role of three different sources of IL-33 were discussed in lines 467-496 of the revised manuscript.

sST2

There are few ST2-positive cells in the control sample and low expression of the transcripts variant 2/3 (encoding sST2) according to the flow cytometry which we presented previously (Fig. 4a and Supplementary Fig. 15), and the qPCR results we presented previously showed few cells producing sST2 when the tissue was intact (Fig. 3c-f and Supplementary Fig. 14a, b). Our immunofluorescence staining results for ST2 in the revised manuscript (Fig. 4g and Supplementary Fig. 17) show that the expression was observed in 1) some CD45⁺ immune cells and 2) few of CD45⁻EpCAM⁻Thy-1.2⁻ fibroblast/stromal cells in PRT of control samples. For the specific type of population 1, we found the mST2⁺ periodontal tissue-resident macrophages (PTRMs) in additional experiments, and this population was stably present in PRT regardless of whether inflammation was induced (Fig. 6a-c, and Supplementary Fig. 24, 25). Considering these data, population 1 consists mainly of the mST2-positive PTRMs, whereas sST2-producing cells were very few and hard to find.

When periodontitis was induced, there were three populations in the periodontal tissues: 1) CD45⁺ cells in GT/PRT; 2) CD45⁻EpCAM⁻Thy-1.2⁺ fibroblast/stromal cells in GT; 3) CD45⁻EpCAM⁻Thy-1.2⁻ fibroblast/stromal cells in PRT (Fig. 4g and Supplementary Fig. 17). Given their low number compared to CD45⁻ cells when ST2-positive cells were detected in flow cytometry (Fig. 4a and Supplementary Fig. 15), the population 1 was mainly the mST2⁺ PTRMs, following a similar rationale as the control samples (Fig. 6a-c and Supplementary Fig. 24, 25). In contrast, the increased populations 2 and 3 were both sources of sST2 according to the flow cytometry of ST2/mST2 (Fig. 4a, b, 6a-c and Supplementary Fig. 15, 24, 25), as well as mild fold changes in the expressions of transcript variants encoding mST2 (Fig. 3c-f and Supplementary Fig. 14a, b). Thus, although few cells produce sST2 in normal condition, CD45⁻EpCAM⁻Thy-1.2⁺ fibroblast/stromal cells in GT and CD45⁻EpCAM⁻Thy-1.2⁻ fibroblast/stromal in PRT would express large amounts of sST2 upon induction of periodontitis, especially the latter. These results were also described in the lines 270-277, 282-301 and 334-356 in revised

manuscript.

mST2

Because other protein-level methods simultaneously detect sST2, mST2 can only be examined by flow cytometry. We have previously shown that mST2-expressing cells are CD11b⁺ myeloid cells or CD3ε⁺ T cells (Supplementary Fig. 24a, b, percentages and numbers are altered because the re-evaluation of the position of gating), and based on these results, we further investigated the primary specific mST2-rich cells in periodontal tissue in additional experiments. We found that most of them were CD11b⁺MHCII⁺ myeloid cells, with few CD11b⁺MHCII⁻ myeloid or CD3ε⁺ T cells in both tissues (Fig. 6a, b and Supplementary Fig. 24). The percentage of mST2⁺CD11b⁺MHCII⁺ myeloid cells did not change significantly in PRT, but increased in GT after induction of periodontitis. Considering that the absolute number of these cells was higher in intact PRT, it is likely that this population consists mainly of tissue-resident immune cells in PRT, such as tissue-resident macrophages (Fig. 6b and Supplementary Fig. 24a, b). To confirm the cell types of mST2⁺ myeloid lineage populations according to the literature, we applied gating strategies to discriminate the major antigen-presenting myeloid cell subsets (APCMC, including dendritic cells, M1/M2 macrophages), non-antigen presenting myeloid cell subsets (non-APCMC, including monocytes, neutrophils, and mast cells), and helper and regulatory T cell subsets (including Th1, Th2, Th17, and Treg; details are documented in the reporting summary). Notably, we clarified that the CD11b⁺MHCII⁺CD11c⁻CD86⁻CD206⁺ macrophage (defined as periodontal tissue-resident macrophages, PTRM) subset was a major antigen-presenting cell population in periodontal tissue, based on previous results showing that APCMCs are abundant in periodontal tissue (Fig. 6c and Supplementary Fig. 24, 25a). Dendritic cells and M1/M2 macrophages also had mST2-positive subpopulations, but relatively few compared to the leading role the PTRMs (Fig. 6c). In contrast, mST2⁺ non-APCMCs were also present, albeit in low absolute numbers, and were relatively enriched in monocytes and mast cells, with few eosinophils or basophils (Supplementary Fig. 25b, c). We also tried to determine the subpopulation of mST2⁺ T cells, but this was difficult due to their scarcity in the local Th and regulatory cell populations (Supplementary Fig. 25d, e). Therefore, mST2⁺ cells are largely consists of the PTRMs.

We thank reviewer #3 again for suggesting this point, which made us consider deeply about the mechanism of the IL-33/ST2 axis. We hope that the reviewer #3 will be satisfied with this additional content mainly described in lines 334-356.

10. In Figure 6a, *Il33* Δ/Δ and *Il1rl1* $^{-/-}$ mice, severe bone loss was shown at Day 8, but this lost bone was apparently restored at Day 14. How can the authors explain this? Does this mean the bone was regenerated or are there different roles for the IL-33/ST2 axis at different stages? What are the cellular changes at Day 14?

In *Il33* Δ/Δ and *Il1rl1* $^{-/-}$ mice, we consider bone regeneration was faster due to lower expression of both genes after Day 8, suggesting that the deficiency of the IL-33/ST2 axis mainly affects the inflammation. (Fig. 3c–h and Supplementary Fig. 14a, b, d–g). To investigate this point, bulk RNA-seq was performed on PRT and BT samples from 11-day ligated WT and *Il33* Δ/Δ mice (the time point represents the process from AP to CP). Pathway analysis between them showed enrichment of suppression of pathways associated with immune reaction in PRT and upregulation of pathways associated with regeneration in BT (Supplementary Fig. 23a–c and Table 3, 4). These results support our view that there is more substantial immunosuppression and bone regeneration in *Il33* Δ/Δ mice (lines 323–332). Although the phenomenon raised by reviewer #3 is important, the specific mechanism for this process could be performed in further studies of the chronic process of periodontitis, and will be the next major research questions to be addressed.

11. The cellular sources of IL-33 and *Il1rl1* as well as their distribution and changes in the three tissues should be detected in *Il33* Δ/Δ and *Il1rl1* $^{-/-}$ mice to verify the importance of the IL-33/ST2 axis.

We examined the changes in *Il33* and *Il1rl1* expression in both two strains of deficient mice, as shown in Reviewer Only Figure 1 below. Several changes were observed, such as the faster decrease of *Il1rl1* when *Il33* was deficient. However, according to our knowledge about the IL-33/ST2 axis, no other receptors or ligands with high affinity for IL-33 or ST2 have been found (Uniprot, <https://www.uniprot.org/uniprotkb/Q8BVZ5/entry>; P14719/entry), and they work in a mutually dependent manner under normal conditions. *Il33* Δ/Δ or *Il1rl1* $^{-/-}$ mice lack the expression of IL-33 or ST2, according to our result and previous studies (Supplementary Fig. 20d; Hoshino, K. et al., J Exp Med, 1999). This fact suggests that ST2 in *Il33* Δ/Δ mice has no ligand to bind or neutralize, and IL-33 in *Il1rl1* $^{-/-}$ mice has no receptor to transduce its signal; thus, the IL-33 or ST2 had little or no effect in *Il1rl1* or *Il33*-deficient mice due to the lack of signal transduction.

Reviewer Only Figure 1 (Fig. R1) Comparison of the expression of *Il33* and *Il1r1* in different tissues between mouse strains. Control; Day 5 of each tissue in WT = 1; $n = 4$.

Minors

1. The authors should annotate the figures to point out the signals or tissues they want to show, for the readers who may be less familiar with periodontitis.

Thank reviewer #3 for the suggestion to improve the readability of our manuscript. We have modified the figure legends or annotated the figure directly to improve the readers' understanding.

2. In Figure 1e, why did the RNA yield increase in the triple ligature model? How does the increased RNA yield support spatiotemporal analysis of the different tissues?

In the single model, the ligature was placed only around the second molar. The area of inflammation was restricted around this tooth, and sampling should also be restricted to this area. However, in the large part of previous studies, the gingiva was collected around all the molars, which ignored the principle of correct sampling. When correctly sampled,

the sample yield was low, especially as shown by the results of PRT (which only sampled around the second molar, Supplementary Fig. 3b). The RNA yield was only about 100 ng in the control PRT of the single model, and this amount is not suitable for RNA-seq analysis. Our triple model allows sampling from all molars by increasing the area of the ligature, which fundamentally improves the RNA yield of PRT, supports the stable RNA-seq analysis, and allows more flexible qPCR analysis from a single individual mouse. Therefore, the increase in RNA yield supported detailed analysis of the different tissues by providing more sample for rare tissue (PRT), allowing more stable analysis and value compared to other tissues (GT and BT). We have added the statement on this point in the methods section (lines 608–610).

3. It would be better to use a marker like CD31 to show angiogenesis.

Please see major point 5 above.

REVIEWER COMMENTS

Reviewer #1 (Remarks to the Author):

The title is still confusing-maybe it can be changed to:

IL-33/ST2 axis involvement in regulation of inflammation in the pathogenesis of periodontitis.

Reviewer #2 (Remarks to the Author):

Manuscript submitted by Liu et al., have significant limitations that would preclude publication in a broad audience journal such as Nature Communications in this reviewer's opinion. While the authors have an interesting and solid finding related to a protective role for IL33/ST2 in experimental periodontitis, the mechanism of action is based largely on correlative data rather than strong mechanistic evidence. Additionally, several observations are not consistent with hypothetical model proposed by the authors. Finally, there are major (English) language issues that render messages either unclear or confusing.

Specifically:

1. The manuscript is difficult to comprehend both because of English language issues as well as confusing conflicting messages related to the data.

Examples of key statements that are unclear or confusing:

Title: "IL33/ST2 axis avoid the excessive inflammation in pathogenesis of periodontitis"

Subheadings: "The initiation of a localized pathology and its relevant systemic alteration"

"Comprehensive analysis that unravels the initiation phase (IP)"

Page 10, lines 166-168 "Given the importance of....using the modified method"

Importantly line 172, periodontitis is not an infection

To mention only a few.

2. The authors were asked by this and other reviewers to investigate the mechanisms of protection from periodontitis in ST2 and IL33 KO mice. While they do additional experiments and add datasets, the mechanism of action is unclear and much of the data does not support their conclusions and is largely confusing. As such:

- The authors show mostly increase of sST2 but then conclude that IL33-mST2 signaling in resident macrophages, shifts macrophages from M2/M1 and drive disease

- The authors conclude that IL33/ST2 signaling is key in macrophage polarization but while they explore macrophage subpopulations in the lesion, they do not explore the (direct or indirect role of IL33/St2) on macrophage polarization

- It is unclear what systemic inflammatory response data suggests

- It is unclear how the authors document increased bone loss without increases in osteoclasts

Reviewer #3 (Remarks to the Author):

All my concerns have been addressed. This manuscript has greatly improved.

Regarding the title, I recommend to change it to

The protective role of IL33/ST2 axis in developing acute inflammation in periodontitis

Responses to reviewers' comments on the Nature Communications Manuscript (NCOMMS-23-11381A)

We sincerely thank the three reviewers for careful inspection of our work again, and we are grateful for their positive comments and helpful suggestions. In response to the remaining issues raised by the reviewers, we have performed additional experiments and carefully revised the manuscript, especially for language editing. We believe that the re-revised manuscript takes into account all the comments and hope that it has been improved to the satisfaction of the editors and reviewers. The major changes from the last version of the manuscripts are described below:

Table R1 Major changes in figure contents since the previous manuscript.

Previous manuscript	Alteration	Location in Re-Revised manuscript
Fig. 6	Moved Fig. 6g to Supplementary Fig. 27	Fig. 6 and Supplementary Fig. 27
Fig. 7	Modified	Fig. 7
Supplementary Fig. 27	Added previous Fig. 6g as 27c Changed previous 27c to 27d	Supplementary Fig. 27
Supplementary Fig. 28	Removed because of the weak relationship with the mechanism investigation	/
Supplementary Fig. 33	Modified previous gating strategy as 33a Added new gating strategy of additional experiments as 33b	Supplementary Fig. 33
Supplementary Table. 1/2	Modified	Supplementary Table. 1/2
Supplementary Table. 3/4	Combined	Supplementary Table. 3
Supplementary Table. 5	Modified	Supplementary Table. 4

Table R2 New contents.

Additional figures	Description about contents
Fig. 6g	Flow cytometry analysis of the effect of IL-33 on macrophage polarization in vitro
Supplementary Fig. 28	Flow cytometry and RT-qPCR analysis of the effect of IL-33 on macrophage polarization
Supplementary Fig. 33b	The gating strategy used in new Supplementary Fig. 28

To avoid confusing the reviewers, we note that all the figure and line numbers in the response letter below are from the **re-revised manuscripts** again; none belong to the last one. Please find below our point-by-point responses to each of the reviewers' comments.

Reviewer #1

We would like to thank the reviewer #1 for the helpful comments on our title. We hope that the revision is satisfactory.

1. The title is still confusing-maybe it can be changed to:

IL-33/ST2 axis involvement in regulation of inflammation in the pathogenesis of periodontitis.

We appreciate reviewer #1's point about the problems with the title of the previous manuscript. As suggested by reviewer #1, our title was still confusing and needed to be changed to a more accurate title based on all of our experiments. Considering the suggestions of the other reviewers on the title of the manuscript, we now change our title to "The protective role of the IL-33/ST2 axis in developing acute inflammation during the pathogenetic process of periodontitis".

Reviewer #2

We are very grateful to reviewer #2 for the comments and constructive suggestions on the mechanism exploration and English writing. We greatly appreciate these suggestions, which helped us to improve the credibility and readability of the manuscript. We hope that the manuscript has been improved to the satisfaction of the reviewer.

1. Manuscript submitted by Liu et al., have significant limitations that would preclude publication in a broad audience journal such as Nature Communications in this reviewer's opinion. While the authors have an interesting and solid finding related to a protective role for IL33/ST2 in experimental periodontitis, the mechanism of action is based largely on correlative data rather than strong mechanistic evidence. Additionally, several observations are not consistent with hypothetical model proposed by the authors. Finally, there are major (English) language issues that render messages either unclear or confusing.

Specifically:

1. The manuscript is difficult to comprehend both because of English language issues as well as confusing conflicting messages related to the data.

Examples of key statements that are unclear or confusing:

Title: "IL33/ST2 axis avoid the excessive inflammation in pathogenesis of periodontitis"

Subheadings: "The initiation of a localized pathology and its relevant systemic alteration"

"Comprehensive analysis that unravels the initiation phase (IP)"

Page 10, lines 166-168 "Given the importance of....using the modified method"

Importantly line 172, periodontitis is not an infection

To mention only a few.

We thank reviewer #2 for acknowledging our findings. As reviewer #2 mainly pointed out in this section, our English writing needs to be improved. Regarding the title, we changed it to "The protective role of the IL-33/ST2 axis in developing acute inflammation during the pathogenetic process of periodontitis" after considering the suggestions of other reviewers. For the subheadings mentioned by reviewer #2, we changed them to "The localized and systemic changes of cellular components/cytokine sources in IP " (line 166) and "Involvement of the IL-33/ST2 axis in the pathogenetic process of periodontitis as revealed by comprehensive analysis" (lines 215–216), respectively.

About the specific sentences mentioned by reviewer #2, we changed them to "After

confirming the distinct separation of the tissues with tissue-specific genes (Supplementary Fig. 3c), we investigated the role of the three tissues in the pathogenetic process by examining the expression of representative cytokines using the modified triple ligature model." (lines 140–143) and "We observed an increase in CD45⁺ immune cells and MHCII⁺ antigen-presenting cells in GT and PRT, but a decrease in BT, suggesting that the immune response to invading bacteria occurs mainly in the former (Supplementary Fig. 7a, b)." (lines 169–172). In addition, we have carefully revised the wording of many other sentences not mentioned by reviewer #2 and highlighted them in red. We hope that these changes have now improved the accuracy of our manuscript enough to meet reviewer #2's criteria for scientific writing.

2. The authors were asked by this and other reviewers to investigate the mechanisms of protection from periodontitis in ST2 and IL33 KO mice. While they do additional experiments and add datasets, the mechanism of action is unclear and much of the data does not support their conclusions and is largely confusing. As such:

- The authors show mostly increase of sST2 but then conclude that IL33-mST2 signaling in resident macrophages, shifts macrophages from M2/M1 and drive disease**
- The authors conclude that IL33/ST2 signaling is key in macrophage polarization but while they explore macrophage subpopulations in the lesion, they do not explore the (direct or indirect role of IL33/St2) on macrophage polarization**
- It is unclear what systemic inflammatory response data suggests**
- It is unclear how the authors document increased bone loss without increases in osteoclasts**

We are grateful for reviewer #2's comments on the mechanistic exploration. These comments have greatly helped to improve the credibility and understandability of our work. We respond to them point by point as follows:

- The authors show mostly increase of sST2 but then conclude that IL33-mST2 signaling in resident macrophages, shifts macrophages from M2/M1 and drive disease**

We apologize for not explaining our logic very clearly in the previous manuscript. As we explained in the introduction, *Il1r1*-deficient mice lack the expression of both isoforms of ST2 and lose both the ability to transduce IL-33/ST2 signaling and to competitively

inhibit IL-33 binding to its receptor. To determine which effect is dominant in periodontitis, we tested *Il33*-deficient mice in parallel. If the phenotypes of *Il1rl1*-deficient and *Il33*-deficient mice are similar, it would imply that the phenotype is induced by the loss of the IL-33/ST2 axis and not by other possibilities. Our results showed consistency in the exacerbation of inflammation in periodontitis in both strains of deficient mice (Fig. 5), suggesting that this axis has a protective effect. Although the highly expressed isoform of ST2 was sST2, as pointed out by reviewer #2, the phenotype of both deficient mice showed protective effects induced by IL-33/mST2 signaling. Therefore, in revised manuscript, we have investigated which types of cells express mST2 and found them to be tissue-resident macrophages. In addition, we briefly analyzed the macrophage polarization in both strains of deficient mice and found a shift toward the M1 side and increased IL-6 expression to exacerbate inflammation, implying that the IL-33/ST2 axis in tissue-resident macrophages affects their function to control inflammation. In this revised manuscript, we have improved the details of the description to make this logic clearer (lines 312–320).

- The authors conclude that IL33/ST2 signaling is key in macrophage polarization but while they explore macrophage subpopulations in the lesion, they do not explore the (direct or indirect role of Il33/St2) on macrophage polarization

We thank reviewer #2 for this critical comment. As reviewer #2 pointed out, there is a lack of direct evidence regarding the effect of the IL-33/ST2 axis on macrophage differentiation. Therefore, we performed additional in vitro macrophage polarization experiments (new Fig. 6g and Supplementary Fig. 28 in this re-revised manuscript). We analyzed the expression of IL-33 receptor in macrophages by RT-qPCR and found that M2a macrophages had the highest mST2 (*Il1rl1* variant 1) expression (Supplementary Fig. 28a). Treatment with rmIL-33 significantly enhanced the expression of CD206 and shifted the proportion of polarized macrophages to the M2 side (Fig. 6g and Supplementary Fig. 28b–d). Considering that *Il1rl1* deficiency abolished the M2a polarization-promoting effect of IL-33, our data directly suggested that the IL-33/ST2 axis transduces the signal that enhances M2a macrophage polarization (Fig. 6g and Supplementary Fig. 28). These results are consistent with previous studies (Michael P Nelson *et al.*, *J Immunol*, 2011; Faas, Maria *et al.*, *Immunity*, 2021). We have added several paragraphs in the revised manuscript (lines 391–408).

- It is unclear what systemic inflammatory response data suggests

We appreciate the insightful comment from reviewer #2 regarding the systemic inflammatory response data. As we stated in our manuscript, periodontal disease is not a simple disease that affects only the periodontal tissues, as it also affects other parts of the body (lines 39–41; 525–535). Although the subject of our work was not periodontal medicine, we would like to provide some evidence for subsequent research on chronicity and periodontal medicine and also for researchers working in this field. The results of the systemic effects of periodontitis showed an immunosuppressive state in immune cells in the blood (Fig. 2g–j), and few active bacterial components in tissues after the initial infection (Fig. R1 as below). These results support our hypothesis that the tissue-resident immunoreactive components (cells and cytokines) are more important in the pathogenesis of periodontitis (lines 46–50). We have changed some details of the description (lines 191–213).

Fig. R1 RT-qPCR analysis of 16s rRNA in GT, PRT, and BT.

Top. 16s rRNA levels on the control side of the three tissues (on Day 1; GT = 1; n = 3 or 4).

Bottom. Temporal changes of 16s rRNA levels in the three tissues (control Day 1 of each tissue = 1; n = 3 or 4). GT: gingival tissue, PRT: peri-root tissue, and BT: bone tissue. D1: Day 1; D3: Day 3; D5: Day 5; D8: Day 8; D14: Day 14. Data are presented as the mean ± SEM. *P < 0.05; **P < 0.01; ***P < 0.001; ****P < 0.0001; ns, P > 0.05; by one-way ANOVA with multiple comparisons via Tukey's test; and two-way ANOVA with multiple comparisons via Šídák's method. The obvious outliers were evaluated and excluded by Grubb's test ($\alpha = 0.05$).

- It is unclear how the authors document increased bone loss without increases in osteoclasts

We would like to thank reviewer #2 for the incisive comments regarding increased bone loss without affecting the number of osteoclasts. As reviewer #2 pointed out, the number of osteoclasts was not increased in mice lacking IL-33 or ST2 (Fig. 5d). RT-qPCR results also supported this point, as the *Tnfsf11/Tnfrsf11b* ratio was unchanged (Fig. 5e). However, the eroded surface was significantly increased instead, suggesting that the loss of IL-33/ST2 does not affect osteoclast differentiation in periodontal tissues, but rather enhances its function (Fig. 5d). This point has also been reported in previous studies, using an orthodontic tooth movement model (Izabella L.A. Lima, et al., *Am J Pathol*, 2015).

According to our results, in which the IL-33/ST2 axis acted mainly on periodontal tissue-resident macrophages and altered the M1/M2 ratio, we believe that the change in inflammatory status caused this phenomenon, as it may affect the source of osteoclasts and their characteristics (Yaron Meirou, et al, *Bone Res*, 2022). We have modified the summary figure (Fig. 7b) and the description of the result in Fig. 5a–f (lines 321–333).

Reviewer #3

We are very grateful to reviewer #3 for the kind comments and suggestions regarding the problem in the title of our manuscript. We hope that the manuscript has been improved to the satisfaction of the reviewer.

1.All my concerns have been addressed. This manuscript has greatly improved.

Regarding the title, I recommend to change it to

The protective role of IL33/ST2 axis in developing acute inflammation in periodontitis.

We thank reviewer #3 again for approving our revision. As suggested by reviewer #3, our title still needed to be changed to be more accurate based on all of our experiments. Considering the suggestions of the other reviewers regarding the title of the manuscript, we now change our title to "The protective role of the IL-33/ST2 axis in developing acute inflammation during the pathogenetic process of periodontitis".

REVIEWERS' COMMENTS

Reviewer #2 (Remarks to the Author):

The authors have made significant improvements to their manuscript. The main issue remaining is need for further -editing for language:

Specifically, although the authors have made efforts to modify multiple statements, the manuscript still will need significant language editing before publication.

A second important point is that:

Although the authors have added additional datasets and have greatly improved their study, direct- in depth mechanistic evidence for the role of mST2/IL33 signaling in periodontitis is still lacking

Specifically, the authors conclude that mST2 is responsible for the protective phenotype in ST2KO mice without direct evidence (ie specific deletion of sST2 or mST2). This is inferred by the fact that these mice mimic IL33 KOs. While this is logical it is not direct evidence and therefore something relevant should be included as part of the discussion/limitations of the study section.

Responses to reviewers' comments on the Nature Communications Manuscript (NCOMMS-23-11381B)

Reviewer #2

We are very grateful to reviewer #2 for the acknowledgement of our revision. We greatly appreciate the additional suggestions, which helped us to further improve the accessibility and accuracy of our manuscript. We hope that the manuscript has been sufficiently improved to the satisfaction of the reviewer.

The authors have made significant improvements to their manuscript. The main issue remaining is need for further -editing for language:

Specifically, although the authors have made efforts to modify multiple statements, the manuscript still will need significant language editing before publication.

We regret any mistakes we may have made in our English writing. Given that we are not native English speakers and that our manuscript has changed significantly during the revision process, we had it edited again by an expert. Now we hope that our manuscript is clear enough for publication.

A second important point is that:

Although the authors have added additional datasets and have greatly improved their study, direct- in depth mechanistic evidence for the role of mST2/IL33 signaling in periodontitis is still lacking

Specifically, the authors conclude that mST2 is responsible for the protective phenotype in ST2KO mice without direct evidence (ie specific deletion of sST2 or mST2). This is inferred by the fact that these mice mimic IL33 KOs. While this is logical it is not direct evidence and therefore something relevant should be included as part of the discussion/limitations of the study section.

Thank you for your valuable comment. As reviewer #2 suggested, it would be informative to examine the periodontitis-induced bone destruction in mice with a specific deletion of sST2 or mST2, if available. However, since the transcription variants of sST2 and mST2 are derived from the same gene and the sequence of sST2 is shared by both transcript variants, specific deletion of sST2 is very challenging. Also, although the transcript

variant of mST2 contains three additional exons at the 3'-terminus and appears to be capable of producing the specific deletion allele, the deletion of these exons may have an impact on the splicing of transcript variants of sST2. This makes the mST2-specific deletion difficult to achieve as well. Therefore, we decided to revise a part of the discussion to explain the need for specific deletion as suggested by reviewer #2. We will do our best to design and test the mST2-specific deletion mice in our future studies.